# Beyond Static Vision: Scene Dynamic Field Unlocks Intuitive Physics Understanding in Multi-modal Large Language Models

**Nanxi Li[1], Xiang Wang[2], Yuanjie Chen[1], Haode Zhang[1], Hong Li[1], Yong-Lu Li[1,3]***
[1]Shanghai Jiao Tong University  [2]Tianjin University  [3]Shanghai Innovation Institute
{andyc_03, yonglu_li}@sjtu.edu.cn

## Abstract

While Multimodal Large Language Models (MLLMs) have demonstrated impressive capabilities in image and video understanding, their ability to comprehend the physical world has become an increasingly important research focus. Despite their improvements, current MLLMs struggle significantly with high-level physics reasoning. In this work, we investigate the first step of physical reasoning, i.e., **intuitive physics understanding**, revealing substantial limitations in understanding the dynamics of continuum objects. To isolate and evaluate this specific capability, we introduce two fundamental benchmark tasks: *Next Frame Selection (NFS)* and *Temporal Coherence Verification (TCV)*. Our experiments demonstrate that even state-of-the-art MLLMs perform poorly on these foundational tasks. To address this limitation, we propose *Scene Dynamic Field (SDF)*, a concise approach that leverages physics simulators within a multi-task fine-tuning framework. SDF substantially improves performance, achieving up to 20.7% gains on fluid tasks while showing strong generalization to unseen physical domains. This work not only highlights a critical gap in current MLLMs but also presents a promising cost-efficient approach for developing more physically grounded MLLMs. Our code and data are available at https://github.com/andylinx/Scene-Dynamic-Field.

## 1 Introduction

Recently, Multimodal Large Language Models (MLLMs) have exhibited great success in image and video understanding Yue et al. (2024); Li et al. (2024e). However, MLLMs still face significant limitations in capturing the intuitive physical dynamics of real-world scenarios Zheng et al. (2024); Chow et al. (2025). These shortcomings stem from the **training recipe** and **data property**. The prevalent approach of treating videos as a sequence of frames processed by image encoders and trained end-to-end fails to adequately capture the low-level dynamics essential for understanding physics Labs (2024). On the other hand, video encoders Wang et al. (2023); Zhao et al. (2024a); Wang et al. (2024d); Bardes et al. are often trained in an unsupervised manner on action-focused datasets Kay et al. (2017); Goyal et al. (2017); Kuehne et al. (2011), which are effective for understanding human-centered activities but lack the dynamics of continuum objects such as liquids, cloth, and other deformable materials.

As shown in Figure 1, existing benchmarks Zheng et al. (2024); Chow et al. (2025) mostly evaluate the *high-level* physical reasoning capacities of multimodal large language models (MLLMs). These frameworks assess multiple capabilities through diverse tasks, including physical property question-answering (QA), predictive and counterfactual inference, and spatial-relational analysis. While such tasks are critical for advancing intuitive physics understanding in MLLMs, they are related to not only physics but also vision, language, common sense, logic, etc.. This has led to an alarming performance gap revealed by empirical findings: MLLMs consistently fail across these benchmarks, often performing only marginally better than random guessing.

---

*Corresponding author.

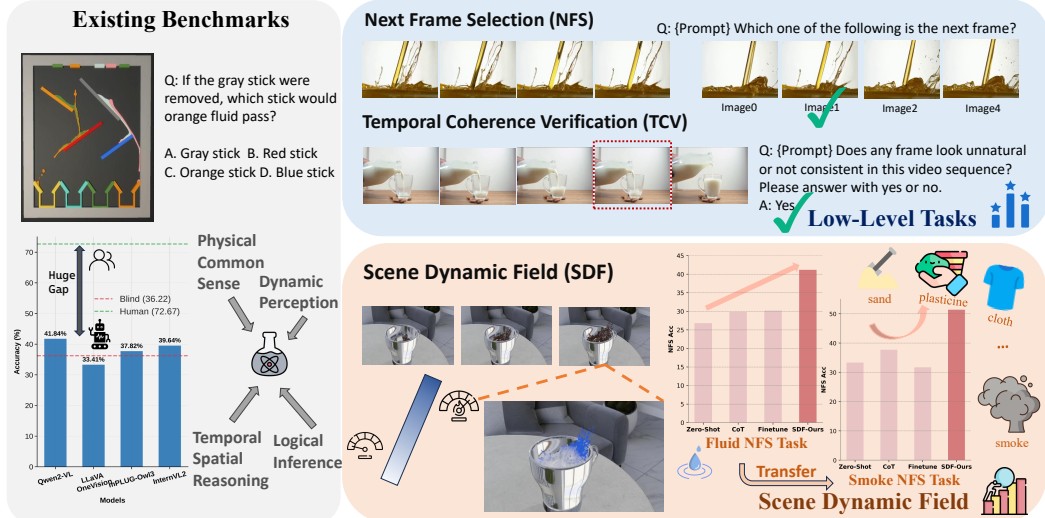

Figure 1: Existing benchmarks entangle multiple capabilities, leading to poor performance in SOTA MLLMs. To address this, we introduce two low-level tasks to assess intuitive physics understanding: Next Frame Selection and Temporal Coherence Verification. Our proposed Scene Dynamic Field (SDF) directly enhances MLLMs' dynamic understanding and shows strong generalization.

Thus, similar to the concept of curriculum learning Bengio et al. (2009), we should also adopt "**curriculum benchmarking**" for MLLM. In this work, we disentangle the physical reasoning benchmarking itself and focus on its very first step, i.e., evaluating the most fundamental and principal physical understanding ability: **intuitive physics understanding**. The reason is that, without establishing whether MLLMs can accurately perceive physical motion and change over time, we cannot meaningfully address their higher-level reasoning deficiencies or develop targeted enhancement strategies. After grounding MLLMs in the fundamental perception of dynamics, we can then enhance their capacity for more complex, causal physical reasoning, as they can operate on a more faithful internal representation of the world's state changes.

It is natural to ask *How can we disentangle the problem to evaluate a model's intuitive physics understanding capability effectively?* and *How can we further enhance this critical ability?* To this end, we introduce two low-level tasks: **Next Frame Selection (NFS)** and **Temporal Coherence Verification (TCV)**, which systematically assess intuitive physics understanding. Extensive experiments reveal that current MLLMs still fall short of achieving a satisfactory level of understanding. Even the best-performing model, Qwen2.5-VL Team (2025), achieves only a $30.0\%$ accuracy on multiple-choice NFS questions. To further investigate these limitations, we analyze the impact of model scaling and find that increasing scale alone cannot lead to significant improvements.

In light of the current reasoning capability of MLLMs, we also explored the potential of language-based reasoning. Despite observing improvement, there remains great potential, since leveraging only language to reason is far from enough to capture real-world physics. Recent progress in **think-with-images** yu Su et al. (2025) motivated us to introduce additional visionary supervisory signals while leveraging existing MLLM capabilities.

To this end, we propose **Scene Dynamic Field (SDF)**, a novel method designed to enhance intuitive physics understanding. SDF leverages physics simulators, which, despite their limitations in fine-grained detail, consistently capture dynamic trends that align with real-world physical phenomena. Through generating SDF data from simulators, we developed a carefully designed multi-task fine-tuning strategy that integrates Chain-of-Thought (CoT) reasoning, where SDF acts as a visual prompt to guide the model's understanding of intuitive physics. Empirical experiments demonstrate significant performance improvements across both benchmark tasks, with up to $20.7\%$ gains on fluid tasks and strong generalization to unseen physical domains, validating the effectiveness of our approach. Moreover, as particle-based physics simulation engines continue to advance in speed and generation quality, our method provides a scalable and cost-effective approach for distilling physical

knowledge from these simulators. By abstracting synthetic data to an optimal representational level, we enable MLLMs to better perceive underlying physical dynamics.

To summarize, our contributions are mainly three-fold:

- We introduce Next Frame Selection (NFS) and Temporal Coherence Verification (TCV), two complementary diagnostic tasks designed to disentangle and evaluate fundamental intuitive physics understanding abilities in MLLMs, revealing their significant deficiencies.
- We propose Scene Dynamic Field (SDF), an intermediate representation that bridges the gap between physics simulators and MLLMs, providing explicit dynamic cues that enhance MLLMs' physical understanding.
- Through extensive empirical analysis, we highlight the significant limitations in intuitive physics understanding of current MLLMs, while demonstrating that our proposed SDF achieves strong generalization, paving the way for more physically grounded MLLMs.

## 2 RELATED WORK

**Intuitive Physics Understanding.** Existing benchmarks for physical reasoning Zheng et al. (2024); Chow et al. (2025) primarily evaluate *high-level* physical reasoning through complex tasks that entangle multiple capabilities. ContPhy Zheng et al. (2024) integrates qualitative reasoning about diverse physical properties with dynamic scenario prediction through video question answering, while PhysBench Chow et al. (2025) proposes a structured evaluation framework containing question-answering tasks across different cognitive levels. These frameworks assess multiple capabilities simultaneously, including vision, language, common sense, logical reasoning, and physical understanding through diverse tasks such as realistic physical simulation Chow et al. (2025); Zheng et al. (2024); Tung et al. (2023), temporal sequence understanding Patel et al. (2022); Tung et al. (2023), and physical property recognition Johnson et al. (2016); Chen et al. (2022); Jassim et al. (2024).

This entanglement has led to concerning empirical findings: studies examining MLLMs' capabilities in intuitive physics understanding reveal that performance approaches random chance levels on such tasks Ballout et al. (2025). Garrido et al. (2025) further confirmed these poor performance findings while revealing that V-JEPA Bardes et al. frameworks show more promise. However, the complexity of existing benchmarks makes it difficult to pinpoint whether models fail due to inadequate physical understanding or deficiencies in other cognitive abilities.

These existing benchmarks leave a fundamental gap: they do not isolate and evaluate **low-level intuitive physical perception**, which is the most basic prerequisite for physical reasoning. Without establishing whether MLLMs can accurately perceive physical motion and change over time, we cannot meaningfully address their higher-level reasoning deficiencies or develop targeted enhancement strategies. Drawing inspiration from curriculum learning principles, we propose a systematic evaluation of this foundational capability through targeted, low-level tasks that isolate physical perception from other cognitive requirements.

Current enhancement approaches have attempted to mitigate these performance issues through various strategies. Some work Sharma et al. (2025) incorporates external tools or components, while other physics-aware AI systems integrate Graph Neural Networks (GNNs) to predict particle dynamics Kazemi et al. (2024) from state vectors or employ symbolic physics engines for explicit reasoning. However, these methods are often designed for multi-step logical deduction rather than empowering foundation models with fundamental perceptual capabilities. Moreover, language-based reasoning improvements, while helpful, prove insufficient for capturing the complex temporal and spatial relationships inherent in physical phenomena.

In contrast, we propose enhancing MLLMs by leveraging physics simulators to provide direct visual cues about physical motion and change, addressing the perceptual limitations at their source rather than relying solely on language-mediated reasoning or external computational modules.

**Multi-modal Large Language Models (MLLMs).** Recently, there has been remarkable progress in MLLMs, with state-of-the-art systems like Qwen-VL series Wang et al. (2024b); Team (2025), InternVL series Chen et al. (2023); Zhu et al. (2025), and GLM-V serires Hong et al. (2025) achieving impressive performance on traditional vision-language benchmarks such as image captioning, visual question answering, and multi-modal reasoning tasks Wang et al. (2024e).

Despite these remarkable achievements, current MLLMs exhibit significant limitations in understanding physical phenomena and dynamics, particularly in non-human-centric scenarios such as fluid dynamics, material deformation, and object interactions governed by physical laws Zheng et al. (2024); Chow et al. (2025). This stems from their training focus on high-level semantic understanding rather than fundamental physical principles. Recent advances have introduced explicit reasoning capabilities through reinforcement learning, with models like GLM-4.1V Hong et al. (2025) achieving success in mathematics and structured reasoning tasks. However, without an accurate perception of physical motion and temporal dynamics, these reasoning mechanisms cannot compensate for the lack of intuitive physics understanding. This creates a critical bottleneck: while current MLLMs excel at complex logical reasoning about abstract concepts, they struggle with basic physical understanding that humans develop intuitively, highlighting the need for systematic evaluation and targeted enhancement of low-level physical perception capabilities.

# 3 BENCHMARK CONSTRUCTION

## 3.1 MOTIVATION

Contemporary video understanding benchmarks Li et al. (2024c); Wang et al. (2024c); Yu et al. (2019) have made significant strides in evaluating high-level event comprehension, successfully capturing human actions Li et al. (2023a; 2019), object interactions, and temporal relationships in videos Li et al. (2024d). However, they often overlook the crucial aspect of low-level dynamic understanding of MLLMs, particularly the intricate physical behaviors and temporal evolution of objects and materials Li et al. (2023b). This limitation is especially apparent when dealing with continuum objects, whose behavior is governed by complex physical principles rather than discrete state changes. Among various continuum phenomena, fluid dynamics presents an ideal testbed due to its ubiquity in everyday scenarios and its rich, continuous dynamic patterns. Therefore, we choose **fluid dynamics** as our primary focus to comprehensively evaluate current foundation models' capability in intuitive dynamics.

To disentangle the problem of physical reasoning, we propose a focused benchmark that specifically evaluates models' capability in intuitive physical understanding, allowing us to disentangle and better comprehend this fundamental intelligence.

## 3.2 EVALUATION SYSTEM

Let $\mathcal{F} = \{f_t\}_{t=1}^{T}$ denote a sequence of $T$ input frames. The strategy proceeds as follows:

**Interval Sampling with Stride**. Partition $\mathcal{F}$ into non-overlapping intervals $\{\mathcal{I}_i\}$ via a temporal stride $s$. Each interval $\mathcal{I}_i = \{f_{t_{\text{start}}}, f_{t_{\text{start}}+1}, \ldots, f_{t_{\text{end}}}\}$ spans a subsequence of frames, where consecutive intervals satisfy $t_{\text{start}}^{(i+1)} - t_{\text{start}}^{(i)} = s$.

**Distractor Candidate Generation**. For each interval $\mathcal{I}_i$, extract a distractor candidate set $\mathcal{D}_i$ by excluding frames within a temporal buffer of size $\delta$ around the interval, then

$$\mathcal{D}_i = \left\{ f_t \,\middle|\, t \notin [\max(1, t_{\text{start}} - \delta), \min(t_{\text{end}} + \delta, T)] \right\}, \tag{1}$$

ensuring $\mathcal{D}_i$ excludes frames that are temporally close to $\mathcal{I}_i$.

**Similarity-Based Pruning**. Let $f_{\text{gt}}$ denote the ground truth frame associated with $\mathcal{I}_i$. Using a similarity metric $\text{sim}(\cdot, \cdot)$, filter out candidates overly aligned with $f_{\text{gt}}$:

$$\mathcal{D}_i' = \left\{ f_t \in \mathcal{D}_i \,\middle|\, \text{sim}(f_t, f_{\text{gt}}) < \tau \right\}, \tag{2}$$

where $\tau$ is a threshold ensuring only semantically distinct distractors are retained. The final evaluation set for $\mathcal{I}_i$ combines $f_{\text{gt}}$ with $\mathcal{D}_i'$, challenging models to isolate the true successor amid plausible alternatives.

We employ two complementary protocols to assess temporal and dynamic understanding capabilities:

**Next Frame Selection (NFS).** For interval $\mathcal{I}_i$ with ground truth $f_{gt}$: First sampling 3 distractors $d_{1:3} \sim \mathcal{D}'_i$, and then computing the selection accuracy $\text{Acc}_{NFS} =$

$$\frac{1}{N} \sum_{i=1}^{N} \mathbb{I}\big(p_{model}(f_{gt}|\mathcal{I}_i) > \max_j p_{model}(d_j|\mathcal{I}_i)\big). \tag{3}$$

| Category | Model | #Param | NFS Acc (%) | | TCV Acc (%) | |
|---|---|---|---|---|---|---|
| | | | **Stride 2** | **Stride 4** | **Stride 2** | **Stride 4** |
| Open-source | InternVL2.5 | 8B | 17.53 | 20.19 | 52.95 | 52.31 |
| | InternVL3 | 8B | 19.33 | 18.33 | 53.00 | 54.94 |
| | GLM-4.1V | 9B | 24.25 | 25.43 | 51.40 | 54.01 |
| | mPLUG-Owl3 | 7B | 27.77 | 29.37 | 52.19 | 53.20 |
| | Qwen2-VL | 7B | 24.00 | 26.80 | 54.00 | 53.24 |
| | Qwen2.5-VL | 7B | 32.73 | 30.00 | 57.80 | 56.63 |
| Closed-source | GPT-4o | — | **33.19** | **39.79** | **70.10** | 69.91 |
| | Gemini-2.5-Flash | — | 31.20 | 31.37 | 67.00 | **70.06** |

Table 1: Performance of models on NFS and TCV tasks. NFS Acc: Accuracy on next frame selection (4-choice MCQ). TCV Acc: Accuracy on temporal coherence verification (Yes/No).

**Temporal Coherence Verification (TCV).** Given sequence $\mathcal{S}$ and corrupted version $\tilde{\mathcal{S}}$ with random distractor insertion, we evaluate binary detection accuracy $\text{Acc}_{TCV} =$

$$\frac{1}{M} \sum_{m=1}^{M} \mathbb{I}\big(\arg\max_{\text{Y/N}} p_{model}(\cdot|\tilde{\mathcal{S}}) = \mathbb{I}(\tilde{\mathcal{S}} = \mathcal{S})\big). \tag{4}$$

This systematic approach mitigates evaluation bias by enforcing diversity in distractor frames while preserving temporal coherence.

### 3.3 DATA PREPARATION

We have adopted fluid video data from multiple sources, including Contphy Zheng et al. (2024) and PhysBench Chow et al. (2025). To enhance the diversity and real-world applicability of our dataset, we supplement these synthetic simulations with **real-world videos** collected through web mining. Specifically, we employ LLMs to generate structured search queries combining action verbs with fluid types (e.g., "pour honey").

To ensure data quality, we segment the collected videos into 5-second clips and implement a robust filtering mechanism. This mechanism utilizes an ensemble of MLLMs, including Qwen-VL-Max Wang et al. (2024b), LLaVA-OneVision Li et al. (2024a), and InternVL2 OpenGVLab Team (2024), to perform consensus-based filtering, eliminating clips that do not contain relevant fluid interactions. Then, we use the above-described process to curate the benchmark. Specifically, we use SigLIP Zhai et al. (2023) embeddings to compute frame-level representations and utilize cosine similarity as our $\text{sim}(\cdot, \cdot)$ metric. To ensure benchmark quality, we incorporate a manual verification phase where human annotators filter out ambiguous or low-quality samples. The final curated test set comprises 4,000 samples derived from about 1,000 unique videos.

### 3.4 ANALYSIS AND INVESTIGATION

We evaluate our benchmark with state-of-the-art MLLMs, including InternVL2.5 OpenGVLab Team (2024), InternVL3 Zhu et al. (2025), mPLUG-Owl3 Ye et al. (2024), Qwen2-VL Wang et al. (2024b), Qwen2.5-VL Team (2025) and GLM4.1V Hong et al. (2025). These models have demonstrated superior performance in various vision-language tasks and feature different model architectures and pre-training strategies. All models are evaluated without any task-specific fine-tuning to assess their zero-shot capabilities. For both the Next Frame Selection (NFS) and Temporal Coherence Verification (TCV) tasks, we conduct experiments with sequences of 5 input frames, utilizing temporal strides of $\delta = 2$ and $\delta = 4$ frames. A detailed ablation is presented in Section B.

| Model | #Param | NFS | TCV |
|---|---|---|---|
| Qwen2-VL | 3B | 24.4 | 51.9 |
|  | 7B | 26.8 | 53.2 |
|  | 72B | 28.1 | 52.1 |
| InternVL2.5 | 2B | 21.3 | 52.9 |
|  | 4B | 22.2 | 51.6 |
|  | 8B | 20.2 | 52.7 |
|  | 26B | 20.6 | 55.1 |

| Model | NFS | TCV |
|---|---|---|
| Qwen2.5-VL | 30.00 | 56.63 |
| + CoT | 31.12 (+1.12) | 64.20 (+7.57) |
| GLM-4.1V | 25.43 | 54.01 |
| + Thinking | 37.20 (+11.77) | 79.07 (+25.06) |
| Gemini-2.5-Flash | 31.37 | 70.06 |
| + Thinking | 37.33 (+5.96) | 72.16 (+2.10) |

Table 2: Model scaling analysis (left) and the effect of CoT prompting and thinking (right).

The experimental results in Table 1 reveal striking deficiencies in current models' intuitive physics understanding capabilities. Open-source models demonstrate particularly poor performance, with most achieving NFS accuracies below 30% (compared to a 25% random baseline for 4-choice selection). Even the best-performing open-source model, Qwen2.5-VL, reaches only 32.73% on NFS. While closed-source models like GPT-4o and Gemini-2.5-Flash show limited improvements, their performance remains far from satisfactory, with NFS accuracies peaking at 39.79%. For the TCV task, open-source models cluster around 52-58% accuracy, only slightly above the 50% random baseline for binary classification. These results collectively indicate that current MLLMs fail to develop effective dynamic representations for intuitive physics understanding.

**Scaling Performance.** Most current foundation models evaluated in our benchmark have parameters around 7-8B, showing limited performance on physical dynamics understanding. To investigate whether this limitation can be addressed through model scaling, we conduct a comprehensive scaling analysis using InternVL2.5 models and Qwen2-VL models on the NFS task.

As shown in Table 2, although larger model sizes generally correlate with better performance, the observed improvements remain relatively incremental. For instance, Qwen2-VL demonstrates progressive gains on NFS, rising from 24.40% to 28.13% as parameters scale from 3B to 72B. However, InternVL2.5 exhibits less consistent scaling behavior, with its NFS accuracy actually declining from 25.33% to 20.60% over the 2B to 26B parameter range. This divergence underscores the limitations of parameter scaling alone for achieving robust comprehension of physical dynamics, suggesting that targeted training methodologies may be necessary to complement pure model size expansion.

**CoT Prompting and Thinking Mode.** As shown in the right panel of Table 2, Chain-of-Thought prompting and thinking modes demonstrate notable improvements over baseline performance. GLM-4.1V benefits most dramatically from thinking mode, achieving gains of 11.77 on NFS and 25.06 on TCV, while CoT provides modest improvements for Qwen2.5-VL (1.12 on NFS, 7.57 on TCV). These results indicate that explicit reasoning processes can enhance intuitive physics understanding, with thinking modes generally outperforming standard CoT approaches across both tasks.

However, while these language-based reasoning improvements are promising, they reveal fundamental limitations in addressing the core challenge of physical dynamics understanding. The gains, though significant, still leave models far from satisfactory performance levels, suggesting that pure linguistic reasoning is insufficient for capturing the complex temporal and spatial relationships inherent in physical phenomena. This observation motivates our approach: rather than relying solely on language-mediated reasoning, we propose to enhance models' perceptual capabilities through explicit visual representations of physical dynamics, leading us to develop the Scene Dynamic Field (SDF) method described in the following section.

## 4 SCENE DYNAMIC FIELD

Our evaluation and comprehensive analysis reveal a significant limitation in current MLLMs: they struggle to effectively understand and reason about physical dynamics. This limitation persists across model scales and architectures, suggesting a fundamental gap in their ability to process temporal dynamics. While language-based reasoning improvements show promise, they are ultimately insufficient for capturing the complex spatio-temporal relationships inherent in physical phenomena.

This observation motivates our approach to enhance the models' perceptual capabilities through explicit visual representations. To this end, we introduce the Scene Dynamic Field (SDF), a lightweight, representation-level bridge that injects physical knowledge without requiring costly architectural overhauls. SDF leverages physics simulators to generate visual prompts that represent motion. Although these simulators may lack fine-grained detail, they consistently capture dynamic trends that align with real-world phenomena. By abstracting these dynamics, SDF provides a valuable foundation for learning, allowing for broad compatibility with existing MLLMs while directly addressing the perceptual gap we identified.

SDF implements an explicit mechanism for modeling temporal evolution and physical interactions through visual prompts derived from physics engines such as Unity Unity Technologies (2023) and Blender Community (2018), thereby complementing MLLMs' inherent strengths in high-level reasoning while addressing their deficiencies in low-level physical understanding.

## 4.1 CONSTRUCT SDF FROM SIMULATOR

We utilized the Flip Fluids Fluids addon, which performed well in Blender Community (2018). To generate simulated videos that contribute meaningfully, we constructed scenes commonly encountered in video-related tasks, such as embodied manipulation and VQA.

**Settings.** We generated a series of videos featuring various liquid-related actions, such as pouring, stirring, and property-comparison demonstrations. To improve the model's generalization across diverse scenarios, we systematically varied multiple factors, including the physical properties of the liquids (e.g., initial velocity, viscosity, and color), the visual characteristics of the containers, the background environments, and the camera perspectives. For details, please refer to the Appendix D.

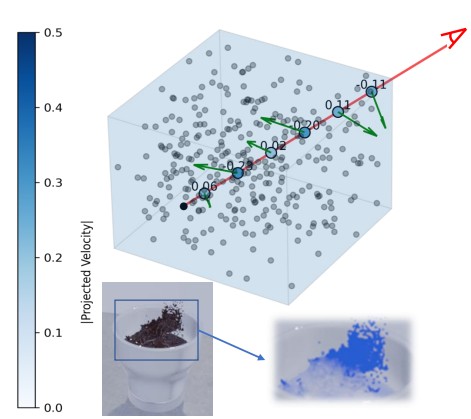

Figure 2: Illustration of our Scene Dynamic Field (SDF).

**Visual Prompting Strategy.** As shown in Figure 2, to enhance the utility of the Scene Dynamic Field for MLLMs, we devised a visual prompting strategy as follows.

Consider a system of particles with velocity vectors $\mathbf{v}_i \in \mathbb{R}^3$. For a camera positioned at $\mathbf{c} \in \mathbb{R}^3$, the projected velocity magnitude $v_{\mathrm{proj},i}$ for each particle is computed as

$$v_{\mathrm{proj},i} = \|\mathbf{v}_i\| \cos\theta_i = (\mathbf{v}_i \cdot \hat{\mathbf{r}}_i), \tag{5}$$

where $\hat{\mathbf{r}}_i = \frac{\mathbf{c}-\mathbf{p}_i}{\|\mathbf{c}-\mathbf{p}_i\|}$ is the unit vector pointing from the particle's position $\mathbf{p}_i$ to the camera. The density of the blue channel $D_B$ is then modeled as a line integral:

$$D_B(\mathbf{c}) = \kappa \int_\Omega \frac{\|\mathbf{v}_i\|}{1 + \alpha\|\mathbf{c}-\mathbf{p}_i\|^2} \, d\Omega, \tag{6}$$

where $\kappa$ scales velocity to color intensity, $\alpha$ governs spatial attenuation, and $\Omega$ represents the observable domain. Particles with larger $v_{\mathrm{proj},i}$ values contribute disproportionately to the blue channel due to the $\|\mathbf{v}_i\|$ term, creating depth perception through spectral segregation. This formulation effectively maps dynamics into the camera's reference frame to a perceptually calibrated blue gradient.

## 4.2 MULTI-TASK FINE-TUNING STRATEGY

Our analysis reveals fundamental gaps in intuitive physics understanding that cannot be resolved through standard video pretraining. We propose a targeted adaptation strategy leveraging physics simulator outputs, structured as follows:

**Task 1: Dynamic Perception**. As depicted in Figure 3, the MLLM is tasked with analyzing an RGB video sequence alongside candidate images to identify the SDF representation, where regions

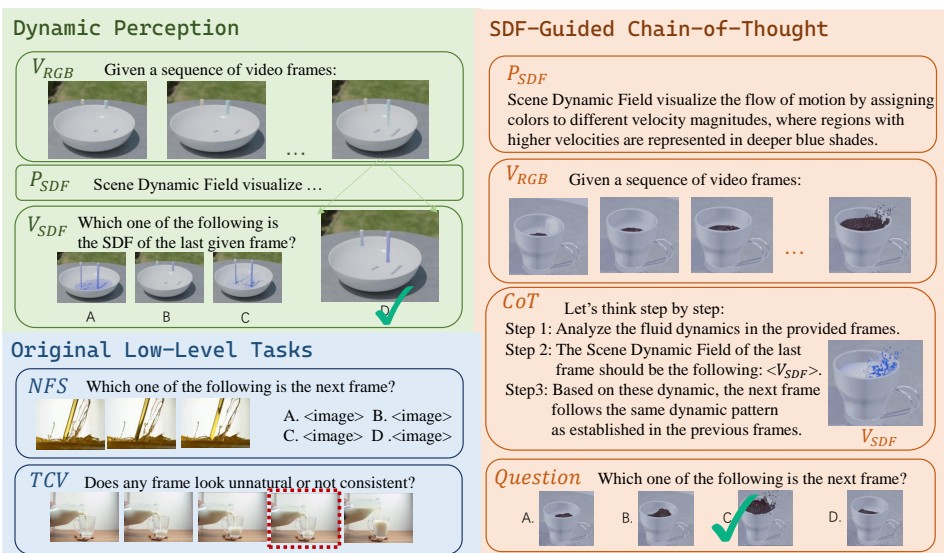

Figure 3: Our multitask framework integrates low-level tasks, a dynamic perception task, and an SDF-guided CoT reasoning task.

of higher velocity magnitude are encoded with increased blue chromatic intensity. The input consists of two components: an RGB video $V_{\text{RGB}} = [I_{\text{RGB}}^1, \ldots, I_{\text{RGB}}^T]$ depicting dynamic interactions, and $N$ candidate images $\{I_1, \ldots, I_N\}$. Among these candidates, one image corresponds to the ground-truth SDF ($I_{\text{SDF}}$) generated through velocity-to-color mapping based on the formulation in Section 4, while the remaining $N - 1$ images serve as distractors.

Following the approach in Section 3.2, we used a similar method to select distractors. These carefully chosen examples balance challenge and clarity, helping the model better understand movement and flow in dynamic scenes.

**Task 2: SDF-Guided Chain-of-Thought**. We develop a three-stage Chain-of-Thought (CoT) reasoning framework enhanced by SDF visual prompts to generate a physics-grounded NFS task.

For input frames $\mathcal{F} = \{f_t\}_{t=1}^T$, insert SDF frames at strategic positions as a visual prompt: $\mathcal{F}_{\text{CoT}} = [f_1^{\text{RGB}}, f_2^{\text{RGB}}, \ldots, f_t^{\text{RGB}}, f_t^{\text{SDF}}]$.

In the context of the NFS, the dynamic field of the final frame plays a critical role in the CoT process. As shown in Figure 3, a meticulously designed three-step CoT framework was developed to enhance dynamic reasoning and predictive capabilities. Initially, MLLM is tasked with analyzing the fluid dynamics present in the provided sequence of frames. Subsequently, it integrates the given scene dynamic field corresponding to the last frame. Finally, the model is required to select the subsequent frame based on the provided visual prompts and reasoned thought processes. This structured approach aims to optimize the model's ability to interpret and predict dynamic scenarios effectively.

Together with Task1, Task2, and the original task (NFS/TCV), we propose to further leverage the model's inherent reasoning capabilities through self-distillation of the thinking procedure. To enhance performance, we incorporate responses from stronger models (e.g., Gemini-2.5-Pro) to guide the effective utilization of our proposed SDF as visual prompts in the reasoning process. However, as demonstrated by recent work Chen et al. (2025), reasoning processes from expert models are not always optimal solutions. Self-distillation approaches can also be promising Zhang et al. (2025), as they exhibit smaller distribution shifts during model training. Therefore, we combine expert-generated data with self-distilled data at a ratio of $1 : 10$. We refer to the Appendix for ablation studies 6 on this choice and the specific prompts employed.

## 4.3 EXPERIMENT

**Settings.** We evaluate four distinct experimental settings on our NFS and TCV benchmarks. The *Zero-Shot* setting evaluates MLLMs without any task-specific tuning to establish baseline performance capabilities. The *Finetune* setting applies supervised fine-tuning to the base model on NFS

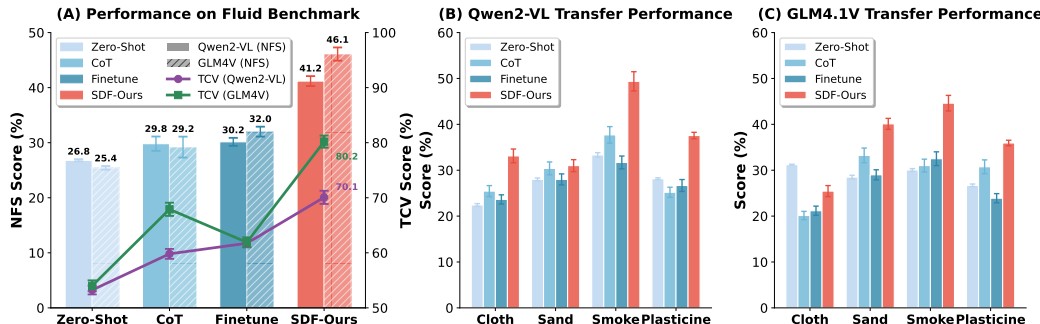

Figure 4: Performance of our SDF method across various evaluation scenarios. (A) shows results on the Fluid dataset for both NFS and TCV tasks. (B) and (C) present transfer results to cloth, smoke, and other particle-based objects on Qwen2-VL and GLM4.1V, respectively.

and TCV tasks separately, using the same number of training instances as our SDF method to ensure a fair comparison. The *CoT (Reasoning Only)* setting employs Chain-of-Thought prompting Wu et al. (2023) without any additional training, leveraging its demonstrated effectiveness in enhancing intuitive physics understanding Jiang et al. (2025). We develop task-specific CoT prompts for both NFS and TCV tasks (see Appendix for details B). Finally, *SDF-Ours* represents our proposed approach, which fine-tunes models using our multi-task framework incorporating Dynamic Perception, SDF-guided CoT, and the original NFS/TCV tasks. Both Finetune and SDF-Ours employ full-parameter supervised fine-tuning via the SWIFT Zhao et al. (2024b) framework, with a learning rate of $1e-5$ across 3 epochs. To show the sim-to-real performance, we test on the previously proposed NFS and TCV datasets, which contain real-world datasets. All experiments are conducted over 5 independent runs on 4 A100 40G GPUs, and results are reported with confidence intervals.

To test the hypothesis that our SDF method facilitates a more fundamental learning of physical dynamics rather than domain-specific memorization, we conducted exploratory transfer experiments on other continuum phenomena, including cloth, sand, and smoke, and other particle-based objects. We conducted transfer experiments on Qwen2-VL and GLM4.1V under identical settings, with both the fine-tuned and SDF configurations trained exclusively on the Fluid dataset. These experiments aim to assess the adaptability and robustness of SDF when applied to diverse physical simulation domains. For more details, please refer to the Appendix 8.

Figure 4 demonstrates the effectiveness of our proposed SDF across various evaluation scenarios. As shown in (A), SDF-Ours achieves substantial improvements on the Fluid benchmark for both NFS and TCV tasks. For the NFS task, SDF-Ours shows a significant performance gain of $14.40\%$ on Qwen2-VL and $20.7\%$ on GLM4.1V compared to Zero-Shot. More remarkably, as illustrated in (B) and (C), while standard Finetune performs nearly identically to Zero-Shot when transferred to other domains (e.g., $23.64\%$ vs. $22.42\%$ for cloth on Qwen2-VL), SDF-Ours maintains noticeable improvements across all transfer domains. These results strongly suggest that our approach facilitates genuine learning of physical dynamics rather than mere domain-specific pattern recognition, enabling effective generalization to diverse physical phenomena beyond the training domain. We provide additional case studies E and failure case analysis G in the Appendix for further discussion.

## 5 CONCLUSION

In this work, we introduced two fundamental tasks for evaluating physical dynamics understanding in multimodal large language models: Next Frame Selection and Temporal Coherence Verification. Through comprehensive experiments, we revealed critical limitations in current MLLMs when it comes to understanding intuitive physics. To address this, we proposed Scene Dynamic Field (SDF), which effectively integrates knowledge from physical simulators into MLLMs. By abstracting physical representations into visual reasoning cues, SDF enables models to better perceive and understand physical dynamics. Our results demonstrate decent improvements across both benchmark tasks, with the method also exhibiting strong transfer capabilities to unseen scenarios.

## 6 ACKNOWLEDGMENT

This work was supported in part by the National Natural Science Foundation of China under Grant No. U25A20442, Shanghai Municipal Science and Technology Major Project No. 2025SHZDZX025G14, National Natural Science Foundation of China under Grant No. 62306175.

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

## APPENDIX

## A  STATEMENT OF LLM USAGE

In the preparation of this manuscript, large language models were used as editorial tools. The LLM assisted in the review of the text to identify potential issues with grammar, spelling, and punctuation. It was also used to find out areas with potential logical inconsistencies or flaws in the presented arguments. All suggestions were reviewed and approved by the human authors, who maintained full control over the final content and voice. The authors take responsibility for the paper's contents.

## B  ABLATION STUDY

**Number of Input Frames.** To investigate the impact of input frame count on model performance, we conducted experiments using Qwen2.5-VL and InternVL2.5. Surprisingly, our results demonstrate that the number of input frames does not significantly affect the model's performance beyond a certain threshold. As shown in Table 3, the performance metrics remain relatively stable across different frame counts, with only marginal improvements when increasing the number of frames. Based on this finding, we adopt 5 input frames as our default configuration for subsequent experiments, as it provides an optimal balance between computational efficiency and model performance.

**Stride.** The selection of temporal stride is a pivotal hyperparameter in our low-level benchmark, as it directly modulates the temporal resolution of frame sampling. As empirically validated in Figure 5, extreme stride values yield suboptimal evaluation regimes. A stride of $\delta = 1$ creates an artificially simplistic task, where models exploit trivial frame-wise similarity cues rather than genuine motion understanding. Conversely, strides exceeding $\delta = 5$ introduce excessive temporal sparsity, causing performance degradation that misrepresents a model's true low-level intuitive physics understanding capabilities due to information loss between sampled frames. To reconcile these competing objectives, we establish $\delta = \{2, 4\}$ as the optimal configuration for our benchmark. This intermediate range ensures sufficient temporal resolution to capture nuanced motion details while maintaining tractable inter-frame variation for robust dynamic understanding evaluation.

| Number of Frames | 3 | 4 | 5 | 6 |
|---|---|---|---|---|
| Qwen2.5-VL | 29.45 | 30.00 | 30.28 | 29.93 |
| InternVL2.5 | 16.80 | 20.19 | 20.27 | 20.18 |

Table 3: Performance analysis of Qwen2.5-VL and Intern2.5VL across different frame counts. Results show minimal variation in key metrics, indicating that the model maintains robust performance regardless of input frame quantity.

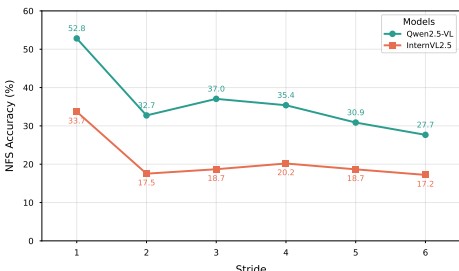

Figure 5: Stride ablation study on the NFS benchmark performance for Qwen2.5-VL and InternVL2.5.

**Prompt Design**

As shown in Table 4, we designed five prompts with different lengths to ablate the performance on the NFS task.

Results in Table 5 demonstrate that different prompts lead to minor performance variations, where prompt complexity exhibits a non-linear relationship with model performance. Interestingly, Prompt

| Index | Prompt Text |
|---|---|
| 1 | Focus on physical dynamics. |
| 2 | Focusing on the physical dynamics within the sequence of video frames is crucial for understanding and solving the problem presented below. |
| 3 | Please carefully pay attention to the physical dynamics, including kinematic properties, dynamic properties, mechanical system properties, fluid dynamics properties. |
| 4 | Understanding the kinematic properties, dynamic properties, mechanical system properties, fluid dynamics properties in the video frame sequence is very helpful to tackle the problem below. |
| 5 | Comprehensively analyze the physical dynamics of the system, with detailed attention to: 1. Kinematic Properties: Displacement, velocity, acceleration, and their relationships for predicting motion within the system. 2. Dynamic Properties: Forces, momentum, and energy conservation. 3. Mechanical System Properties: Friction, elasticity, and damping effects. 4. Fluid Dynamics Properties: Pressure, viscosity, Reynolds number, and flow characteristics. Additionally, ensure both theoretical principles (such as governing equations) and practical applications (e.g., motion prediction or flow behavior) are covered. |

Table 4: The prompts used in the ablation study.

| Prompt Index | 1 | 2 | 3 | 4 | 5 |
|---|---|---|---|---|---|
| Qwen2.5-VL | 28.13 | 29.10 | 30.27 | 29.40 | 27.44 |
| InternVL2.5 | 20.05 | 20.75 | 20.15 | 19.87 | 19.00 |

Table 5: NFS Task performance of Qwen2.5-VL and Intern2.5VL across different prompts. Prompt Index herein corresponds one-to-one with the index in the Table 4 above.

3, which provides explicit guidance on specific physical properties to consider without overwhelming detail, yields the best performance for Qwen2.5-VL (30.27%). In contrast, more verbose prompts (e.g., Prompt 5) or overly simplistic ones (e.g., Prompt 1) show reduced effectiveness. This pattern suggests that MLLMs benefit from focused prompting that directs attention to relevant physical aspects without excessive elaboration. The consistent performance ranking across prompts between models indicates that prompt engineering alone cannot compensate for fundamental model limitations in intuitive physics understanding. This finding reinforces the need for our proposed SDF approach, which addresses these limitations at a more fundamental representational level.

**Expert and Self-distilled Data Mixture Ratio** To better understand the optimal balance between expert data from Gemini-2.5-Pro and self-distilled data, we conduct an ablation study examining model performance across different mixture ratios. We evaluate Qwen2-VL and GLM4.1V as our target models, performing full-parameter supervised fine-tuning with a total of 3,000 data points. Table 6 presents the NFS accuracy under different data mixture ratios. We employ consistent training configurations across all experiments, training for 3 epochs with a learning rate of $1 \times 10^{-5}$ on 4 A100 (40GB) GPUs. The results demonstrate that the optimal balance is achieved at the $1 : 10$ ratio, as excessive expert data can interfere with the model's inherent reasoning processes, while relying

| Model | Expert:Self Data Ratio | | | | | |
|---|---|---|---|---|---|---|
| | 1:0 (Expert) | 1:1 | 1:5 | 1:10 | 1:15 | 0:1 (Self) |
| Qwen2-VL | 32.91 | 38.10 | 39.10 | 41.18 | 39.20 | 31.00 |
| GLM4.1V | 39.91 | 40.70 | 45.90 | 46.11 | 40.20 | 41.00 |

Table 6: NFS accuracy (%) across different expert-to-self-distilled data mixture ratios.

solely on self-distilled data fails to adequately leverage the guidance provided by our proposed SDF framework. Thus, we adopt the $1:10$ ratio as our default setting in all the main experiments.

**Different Representations.**

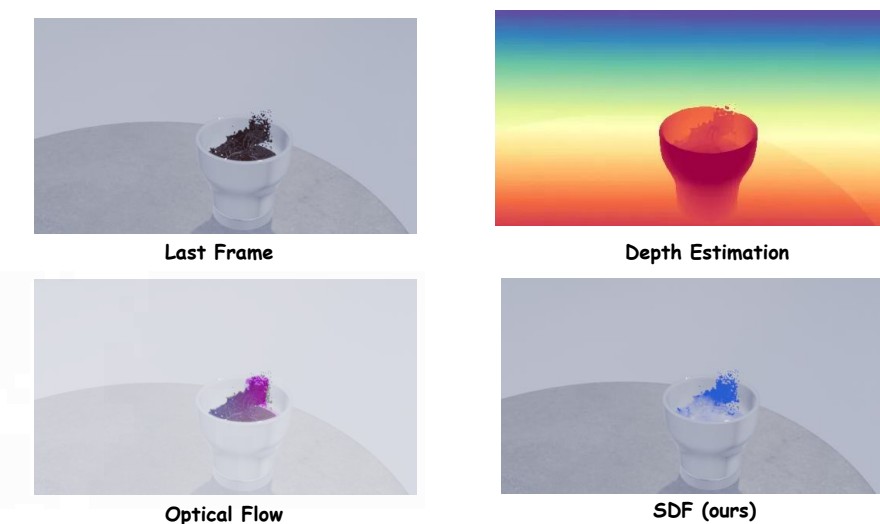

Figure 6: A demonstration of different representations.

Firstly, we argue that reconstructing motion representations from videos such as optical flow is unnecessary for our setting. Our generated simulator provides precise velocity for each particle at every time step. This velocity is a natural source for the Scene Dynamic Field and yields a clean and physically grounded visual prompt for training. In contrast, reconstruction from real world videos is noisy due to occlusion, texture aliasing, camera compression and illumination changes, which weakens the learning signal for intuitive physics.

However, to ablate on the effectiveness of different visual representations, we still conducted an additional comparison across four prompt conditions using the same question answering protocol. To be precise, we randomly select a subset of NFS and TCV tasks and generates its corresponding representations inserted the visual prompt into question as shown in Figure 6.

1. **w/o Visual Prompt:** Direct QA without any visual prompt.
2. **w/ Optical Flow:** Adding the reconstructed optical flow Xu et al. (2022) in the prompt for QA.
3. **w/ Depth:** Adding depth estimation from Depth Anything 3 Lin et al. (2025) of the last frame in the prompt for QA.
4. **w/ SDF (Ours):** Adding our proposed Scene Dynamic Field as a visual prompt for QA.

These results in Table 7 indicate that while optical flow provides some helpful motion cues for the Temporal Coherence Verification (TCV) task, it is less effective on Next Frame Selection (NFS), where reconstruction noise likely limits its discriminative value. In contrast, SDF achieves the best

Table 7: Ablation study comparing the effectiveness of different visual prompts on NFS and TCV.

| Visual Prompt | NFS Acc (%) | TCV Acc (%) |
|---|---|---|
| w/o Visual Prompt | 29.1 | 57.5 |
| w/ Optical Flow | 27.0 | 65.1 |
| w/ Depth | 32.4 | 60.1 |
| **w/ SDF (Ours)** | **41.2** | **68.9** |

performance across both tasks, demonstrating that clean velocity data derived from the simulator yields a more reliable and generalizable visual prompt.

## C    DATA PREPARATION PIPELINE

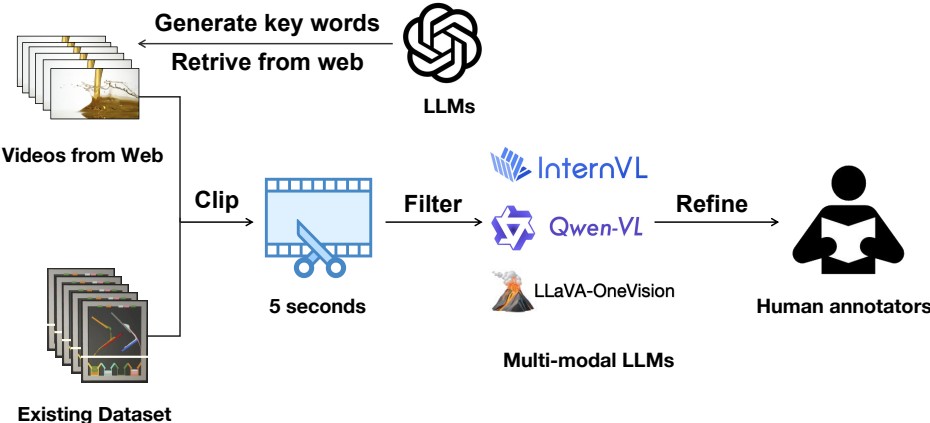

Figure 7: A demonstration of our data preparation pipeline.

We have architected and deployed a data preparation pipeline, as shown in Figure 7. We have curated our fluid video dataset by aggregating synthetic simulation data from established repositories such as Contphy Zheng et al. (2024) and PhysBench Chow et al. (2025), complemented by an extensive collection of real-world scenario videos featuring liquid-related actions through web mining. During implementation, we employ Large Language Models(LLMs) to generate retrieval keywords. The keywords are mainly composed of two parts: actions (e.g., "pour", "stir", "shake") and categories of liquids (e.g., "oil", "milk", "juice"), with each action paired with a liquid category to form a search query(e.g., "pour oil", "pour juice", "stir milk"). To ensure data quality, we segmented the collected video data into 5-second clips and employed an ensemble of multimodal LLMs, including Qwen-VL-Max Wang et al. (2024b), LLaVA-OneVision Li et al. (2024a), and InternVL2 OpenGVLab Team (2024), to filter out those clips that do not contain liquid interactions. For data refinement, we continue to develop a filtering interface for human annotators, as shown in Figure 8, which further filters the videos with noticeable perspective shifts or obvious changes in playback speed from the human perspective, ensuring that the content remains consistent and coherent. For potential ethical concerns, this user interface incorporates dedicated options for annotators to report issues related to privacy or other ethical considerations.

## D    SCENE DYNAMIC FIELD SETTINGS

To improve generalization, we construct scenes that encompass a variety of liquid-related actions, systematically varying the physical properties of the liquid, the visual attributes of the containers, the

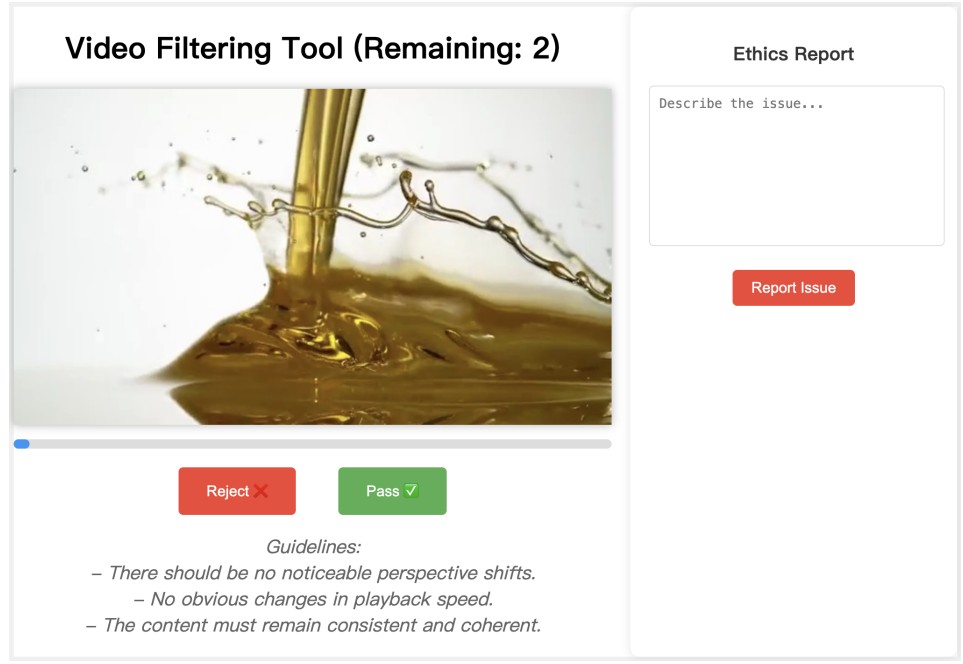

Figure 8: The human evaluation interface in data refinement is designed to filter low-quality data and flag potential ethical issues.

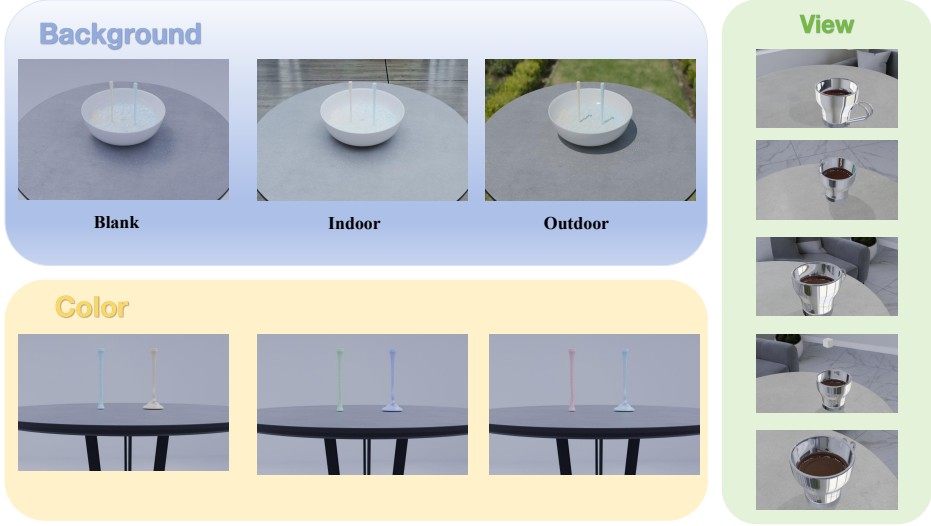

Figure 9: A demonstration of various environment backgrounds, liquid color, viscosity, and camera views settings.

background environments, and the camera perspectives. During implementation, we selected three representative viscosity values within an appropriate range, corresponding to liquids with low viscosity (e.g., water), moderate viscosity (e.g., oil), and high viscosity (e.g., honey). Additionally, we introduced variations in liquid color to ensure a diverse visual representation by altering the materials of the fluid_surface in FLIPMeshes. Furthermore, we incorporated three representative background environments—blank, indoor, and outdoor—along with five distinct camera perspectives to enhance the robustness of our model across varying physical contexts. This was achieved by modifying the environment HDRs and adding cameras from different angles in Blender. A demonstration of these different settings is shown in Figure 9.

# E CASE STUDY

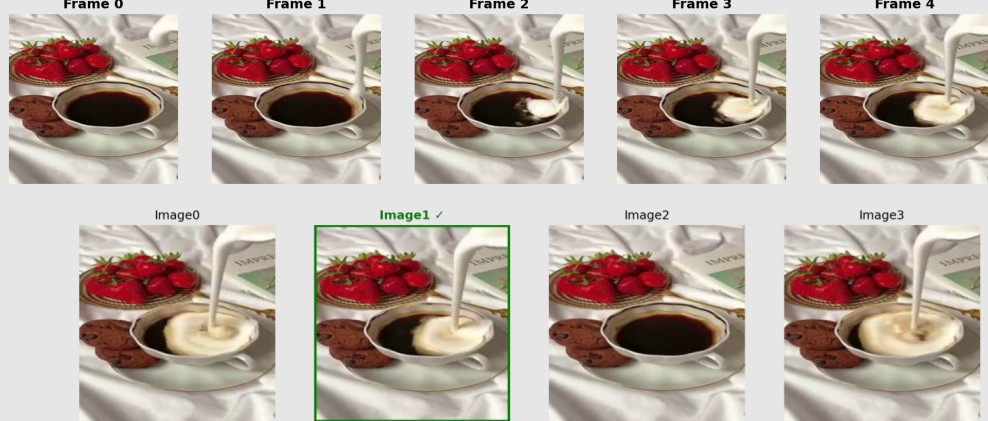

**Input**

Given a sequence of video frames [frame list]. Which one of the following four images is more likely to be the next frame? Your answer should be in the format of Conclusion: [Image0 or Image1 or Image2 or Image3].

**Qwen2-VL**

Conclusion: Image 0. 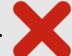

**Qwen2-VL + SDF (ours)**

To determine which image is most likely to be the next frame, let's analyze the sequence of actions in the frames:
**Frame 0 to Frame 4**: Milk is poured with a steady velocity (consistent flow rate) and a downward direction into the coffee cup. In Frame 4, the milk-pouring action is still ongoing (the stream of milk is visible).
**Evaluate the options**:
Image0: Shows the pouring action has stopped, with a stable liquid level. This does not match the ongoing action in Frame 4.
Image1: Shows the pouring action is still ongoing, with the liquid level rising**. Retains the steady pouring velocity and downward direction of milk into the cup, consistent with the continuous action in Frame 4.**
Image2: Shows the pouring action has stopped, with a stable liquid level. This does not match the ongoing action.
Image3: Shows the pouring action has stopped, with a stable liquid level. This does not match the ongoing action.
Since the pouring action is still in progress in Frame 4, the next frame should continue this action, which aligns with Image1.

Conclusion: Image1. ✓

Figure 10: Case study on Qwen2-VL for zero-shot and SDF setting.

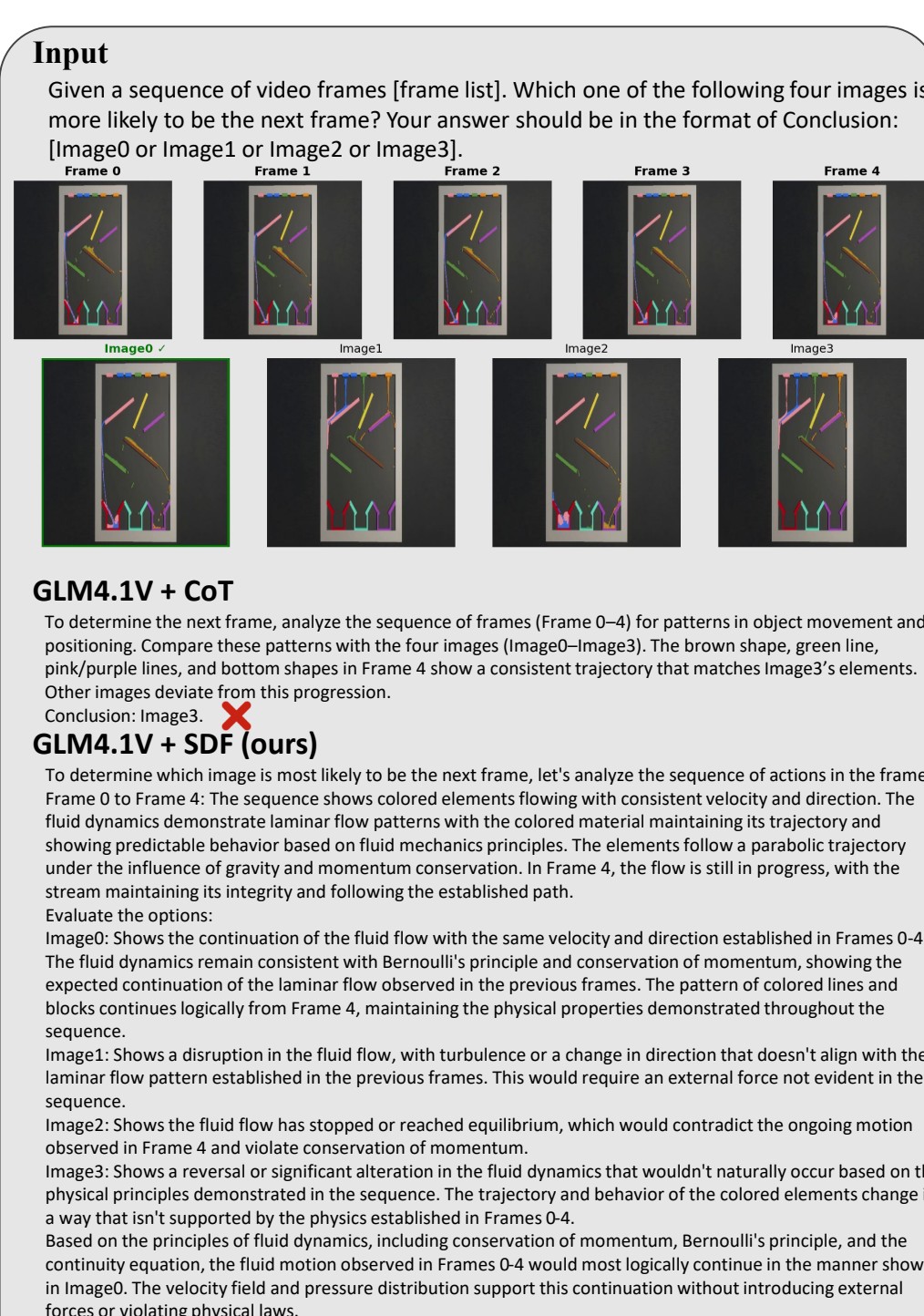

Figure 11: Case study on GLM4.1V for CoT and SDF setting.

As demonstrated in Figure 10, our proposed SDF method successfully enhances the Qwen2-VL model's ability to reason about the spatial and physical dynamics of continuum objects. The method enables the model to generate physically grounded interpretations, such as recognizing that milk maintains a steady pouring velocity and follows a downward trajectory into the cup. This exem-

plifies how SDF can effectively inject physical visual perception and reasoning capabilities into the model. Furthermore, Figure 11 illustrates a more challenging scenario where our SDF-trained GLM4.1V tackles a complex problem from Contphy Zheng et al. (2024) involving multiple simultaneous fluid flows. Our trained model demonstrates the ability to compare and reason about velocity and momentum characteristics to arrive at the correct answer. When we directly prompt the model to think step-by-step, the CoT may be influenced by superficial visual features such as color or appearance rather than relying purely on deeper physical knowledge.

## F    IMPLEMENTATION DETAILS

### F.1    CoT PROMPTS

Chain-of-Thought prompting can enhance the reasoning capabilities of Multi-modal Large Language Models (MLLMs) in tasks such as Visual Question Answering (VQA) Wu et al. (2023). By explicitly guiding the model through a series of intermediate reasoning steps, CoT enables a deeper understanding and more accurate responses, particularly when complex visual or multimodal information is involved. To demonstrate this, we have designed distinct CoT prompts tailored to both the NFS (Next Frame Selection) and TCV (Temporal Coherence Verification) tasks, optimizing the models' ability to reason through intricate queries and generate more informed answers.

**CoT prompt for NFS task:**
*Let's think step by step. To determine which of the four images is most likely to be the next frame in the sequence, we can approach the task by analyzing the sequence of frames and considering the following chain of thought: Sequence Analysis: Look at the sequence of video frames from Frame-0 to Frame-4. Assess the general movement, objects, or changes between these frames. Identify any objects or actions that are progressively changing, such as motion, appearance, or position. Motion and Trends: Identify patterns or trends within the sequence: whether any objects are moving in a consistent direction or interacting with other objects. This can help to predict the direction or type of changes to expect in the next frame. Consistency with Previous Frames: Compare the four candidate images (Image0, Image1, Image2, Image3) with the sequence. Which one follows the motion, appearance, or transitions observed in the previous frames? Does one of the candidate images exhibit the next logical state or movement? Discarding Outliers: If any of the candidate images significantly diverges from the observed trends in the sequence (e.g., in terms of position, motion, or object changes), it can be ruled out as less likely. Final Prediction: Based on the trends and the consistency of the images with the observed dynamics, choose the most plausible candidate image. Now, apply these steps to the sequence of frames and the candidate images, and determine which one is most likely to be the next frame.*

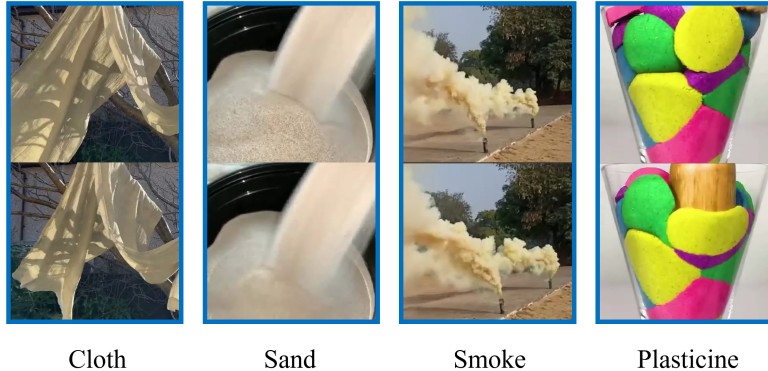

| Cloth | Sand | Smoke | Plasticine |

Figure 12: The demo of the data we collected and used in transfer experiments. In each field of the image, the upper and lower frames are sampled in the same 3-second clip, and the lower frame is later than the upper frame.

**CoT prompt for TCV task:**
*Let's think step by step. Inspect the video frames one by one for obvious irregularities; compare object movement and appearance for consistency across frames; check for any unusual transitions, lighting, or distortions; assess background and environmental consistency; if any frame diverges in a way that violates natural progression, answer "yes", otherwise answer "no".*

**Annotation prompt for self-annotating process:**
*You are an expert in visual reasoning. Your task is to generate a step-by-step "thought process" that logically explains how to arrive at the correct answer for a given question. You will be provided with the question and its ground truth answer. Your output should only be the reasoning steps. Do not repeat the question; just generate the thinking process. [START OF TASK] 1. QUESTION: {question} 2. GROUND TRUTH ANSWER: {gt choice} You should end with "Conclusion: {gt choice}".*

**Annotation prompt for Gemini-2.5-pro:**
*You are an expert in visual reasoning. Your task is to generate a step-by-step "thought process" that logically explains how to arrive at the correct answer for a given question. You will be provided with the question and its ground truth answer. Your output should only be the reasoning steps. Do not repeat the question; just generate the thinking process. [START OF TASK] 1. QUESTION: Given a sequence of video frames: [Video Frames] Scene Dynamic Field (SDF) visualizes the flow of motion by assigning colors to different velocity magnitudes, where regions with higher velocities are represented in deeper blue shades. Which one of the following is the SDF of the last given frame? [Choices] Your response should be one of the following: A, B, C, D. 2. GROUND TRUTH ANSWER: {gt choice}*

# G   FAILURE CASES ANALYSIS

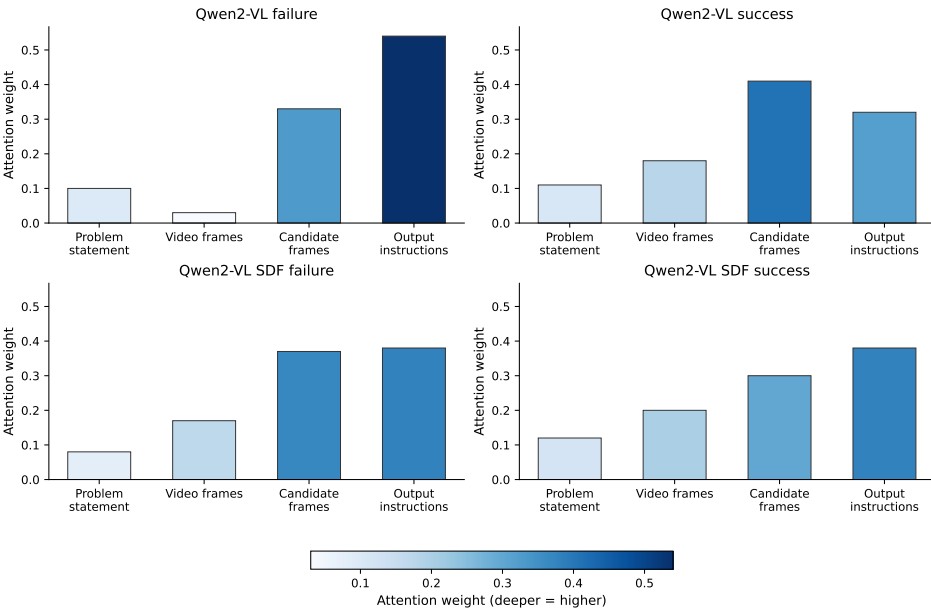

Figure 13: Attention weight visualization for Qwen2-VL on the NFS task. Left panels show successful cases, right panels show failed cases. Top row: zero-shot setting; bottom row: our proposed SDF setting. Darker colors indicate higher attention weights.

Different from high-level physical reasoning tasks, which can be further disentangled into several sub-steps, fundamental physical understanding tasks represent low-level perception capabilities that are often harder to probe. Therefore, we quantify the attention weights for model inputs without explicitly requiring output reasoning processes. As found by previous work Li et al. (2024b); Wang et al. (2024a), MLLMs are weak at long-context reasoning, especially when the input consists of multiple images. This similar phenomenon is also observed in our experiments.

As shown in Figure 13, we average the attention weights across all output tokens and all layers to analyze the model's focus distribution. The attention weights of the output instruction and candidate frames part are significantly higher than those of the previous video frames in the zero-shot setting. This indicates that the model tends to focus on the most recent frames, potentially overlooking important temporal dynamics present in earlier frames. This bias towards recent information can lead to suboptimal performance in tasks that require a comprehensive understanding of the entire sequence.

We can observe higher attention weights on earlier video frames in successful samples compared to failed ones. The failed cases often focus more on choices and instruction parts, failing to attend to the equally important video frames. This suggests that the model may be overly influenced by the provided choices and instructions, rather than fully engaging with the visual content where continuum dynamics (e.g., fluid motion) are most apparent. Importantly, our proposed SDF method demonstrates a clear improvement in attention distribution. As illustrated in the bottom row of Figure 13, SDF, to some extent, draws more attention to the equally important video frames where continuum phenomena such as fluid dynamics can be observed. This enhanced attention allocation enables the model to better capture the temporal evolution of physical processes and make more informed predictions, ultimately leading to improved performance in selecting the correct choice.

Examining the explicit reasoning processes reveals additional insights into failure mechanisms. We observe that visual hallucination becomes more prevalent in longer reasoning sequences, consistent with findings from prior hallucination research Liu et al. (2025). Our analysis shows that extended reasoning leads to approximately 31% reduced attention allocation to visual tokens, thereby diminishing perceptual capabilities. This attention degradation is particularly pronounced in later stages of the reasoning process, where models increasingly rely on internally generated content rather than grounding their responses in the provided visual information.

In all, these failure cases show the fundamental challenges MLLMs encounter when integrating and reasoning over temporal visual sequences. The observed patterns highlight the critical need for enhanced attention mechanisms and reasoning frameworks that can more effectively leverage the complete contextual information within visual sequences, particularly for tasks involving complex physical dynamics. These findings suggest that future research should pay more attention to developing more supervision mechanisms for visual perception capabilities throughout the reasoning process. Additionally, incorporating physically grounded training data will be essential for cultivating more physics-aware MLLMs Polverini & Gregorcic (2025), ultimately enabling their broader application across diverse physical reasoning domains Gao et al. (2023).

## H  ENCODER EVALUATION

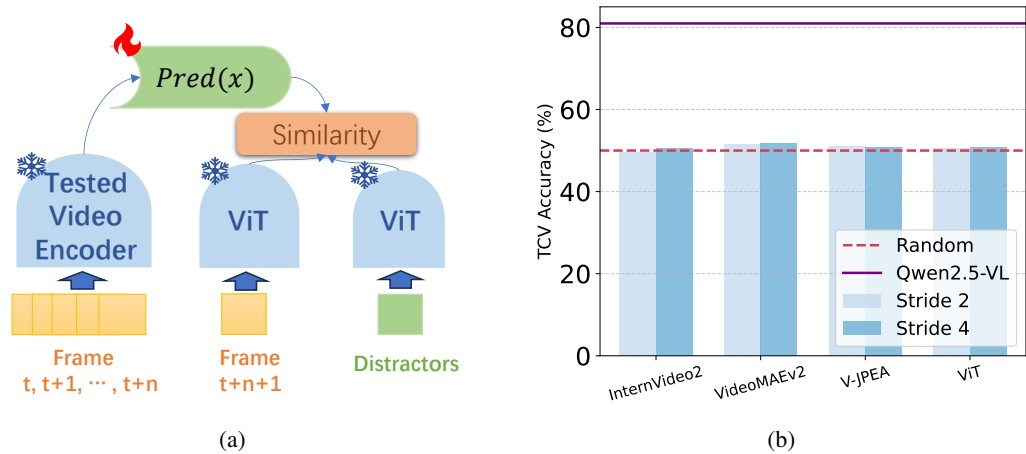

(a)                                                                        (b)

Figure 14: (a) Architectural illustration of the V-JEPA-like encoder test. (b) Performance evaluation of various video encoders on TCV tasks.

Given the relatively poor performance of current foundation models on physical dynamics understanding, a natural question arises: can we obtain better performance by effectively leveraging the encoders? To investigate this possibility, we designed an experiment inspired by the V-JEPA Bardes et al. (2024) architecture to directly test the representational power of various encoders.

We employ a three-layer Transformer-based prediction network (Pred(x)) that takes as input the representation generated by a frozen video encoder. This encoder processes a sequence of consecutive video frames — specifically, frames t - t+n — to produce a compact, contextualized embedding.

The output of the prediction network is then compared via a similarity function (e.g., cosine similarity) against a fixed reference representation derived from a pre-trained Vision Transformer (ViT). This reference ViT encodes a single target frame (frame t+n+1) representing the immediate future frame.

Additionally, the same similarity metric is computed between the predicted representation and embeddings of "distractor" frames (also encoded by the same frozen ViT), which serve as negative examples during training. This contrastive setup encourages the prediction network to learn representations that are semantically aligned with the true future frame while being distinct from irrelevant distractors.

The entire system is trained end-to-end to maximize the similarity between predicted and target representations, while minimizing similarity with distractors — effectively learning to anticipate visual content in a self-supervised manner.

Upon evaluation on a simple subset of TCV tasks in Figure 14, it was observed that directly leveraging the visual representation alone was insufficient for capturing the complexities of dynamic behavior, as evidenced by a significant performance gap compared to the Qwen2.5-VL model. It motivates further exploration into harnessing the inherent capabilities of MLLMs to enhance the representational power needed for intuitive physics understanding.

This finding led us to focus on the data-centric perspective. The recent improvements in models like Qwen3-VL, which was trained on more embodied and spatial task data, lend strong support to this view. It's performance improved on our intuitive physical understanding tasks with the help of these spatial and physical data. It indicates that the underlying architectures of MLLMs are capable of learning physical reasoning, but they must be exposed to the right kind of data. We believe that intuitive physical understanding is a foundational capability. Therefore, attempting to learn it implicitly from high level, end to end task data (such as embodied action) would be akin to putting the cart before the horse. A more effective strategy is to first equip MLLMs with this fundamental capability using targeted data.

Therefore, our work adopts a data centric approach precisely to address this gap. By generating structured data focused on physical dynamics (SDF), we aim to directly enhance the MLLM's capacity for intuitive physical reasoning, providing a solid foundation for a wide range of real world applications.

# I    TRANSFER EXPERIMENT DETAILS

| Domain | Object | Action |
|--------|--------|--------|
| Cloth | cloth, skirt, shirt, ribbon | fluttering |
| Sand | sand, grain, lime, seeds | pouring, falling |
| Smoke | steam, smoke | ejecting, spreading |
| Plasticine | plasticine, kinetic sand | pouring, cutting, stretching |

Table 8: The data composition for transfer experiments.

During experiments within the liquid domain, we found that MLLMs can accurately predict the deformation of liquids by capturing the viscosity of fluid particles, cohesive forces, and the laws of momentum transfer. This suggests that MLLMs may have developed the capability to model the "dynamics of continuous objects formed by particle aggregates." Fluids share deep-seated similarities in physical essence with materials such as cloth, sand, smoke, and plasticine. Traditional

methods require customized physics engines to model these materials, whereas the generalization ability of MLLM may break through domain barriers and achieve unified representation. Therefore, we adopted the same paradigm (shown in Section C) used for liquid dataset collection, as shown in Figure 12, to gather data from four domains: cloth, sand, smoke, and plasticine. Table 8 shows the composition of the data.

We acknowledge that our study primarily centers on fluid dynamics as a representative case for continuum physics. And our exploratory transfer experiments show promising generalization to other particle-based materials. We do not exhaust the full spectrum of intuitive physics, which also includes principles governing rigid body collisions, thermodynamics, or optics, since fluid has always been proposed to be a difficult Zheng et al. (2024) but useful material to start with (e.g., robotics manipulation tasks Lin et al. (2020)). Future work should develop more comprehensive benchmarks that span a wider array of physical phenomena to fully assess and cultivate a holistic physical understanding in MLLMs.

