# OpenReview forum: "Beyond Static Vision: Scene Dynamic Field Unlocks Intuitive Physics Understanding in Multi-modal Large Language Models"
_ICLR.cc/2026/Conference — ICLR 2026 Poster_

### Official Review · Reviewer_HQBD · 2025-10-16

**Soundness:** 3
**Presentation:** 3
**Contribution:** 3
**Rating:** 6
**Confidence:** 4

**Summary:**

This paper addresses a critical limitation in MLLMs: their poor understanding of intuitive physics, particularly for continuum objects like fluids. The authors introduce two diagnostic tasks—Next Frame Selection (NFS) and Temporal Coherence Verification (TCV)—to systematically evaluate low-level physical perception. They reveal that even state-of-the-art MLLMs perform poorly on these tasks, often near random baselines. To bridge this gap, the authors propose Scene Dynamic Field (SDF), a method that leverages physics simulators to generate visual prompts representing motion dynamics. Through multi-task fine-tuning, SDF significantly improves model performance and generalizes well to unseen physical domains like cloth and smoke.

**Strengths:**

1. The paper is interesting for addressing the problem of poor low-level physical perception by proposing practical physics simulators for visual prompting to improve performance.

2. The paper is well-organized and accessible, with clear explanations of both the problem and the solution.

**Weaknesses:**

1. Limited Scope of Transfer Experiments: While the transfer experiments across continuum domains (cloth, sand, smoke, plasticine) are promising, the scope remains limited. The paper would be significantly strengthened by evaluating the method's generalization to other fundamental physical phenomena, such as rigid-body dynamics, collisions, or optical effects.

2. Experimental Design and Baselines: To provide a more comprehensive performance comparison, we suggest expanding Table 1 to include additional baseline models. Specifically, it would be informative to compare against larger models with more parameters.

3. Dependence on Synthetic Data and Real-World Applicability: The SDF method's reliance on synthetic data from simulators is a potential limitation, as such data may not fully capture the complexity and noise of real-world physical systems. The paper could be improved by discussing the feasibility and potential challenges of applying this method to real-world physics problems. For instance, how would the SDF approach perform with real sensor data that is often incomplete or noisy?

4. Analysis of MLLM Limitations: The paper identifies that MLLMs struggle with low-level dynamics but does not deeply investigate the root cause. A more thorough analysis is needed to determine whether these failures stem from architectural limitations (e.g., an inductive bias towards high-level semantics) or a bias in the pre-training data (e.g., a lack of low-level physical reasoning examples). Uncovering this would provide valuable insight for future research.

**Questions:**

See weaknesses. I am willing to discuss with the authors

---

> ### Author Response · Authors · 2025-11-20
> **Rebuttal to Reviewer HQBD (Part 1/3)**
>
> We sincerely thank the reviewer for their constructive feedback. We are encouraged that they found our work "interesting" and "well-organized," and our use of physics simulators "practical." We will address each of the reviewer's valuable suggestions below.
>
> ### Q1. Scope of Transfer Experiments
>
> We agree that our current transfer experiments focus on continuum dynamics. Our hypothesis was to first tackle particle-based dynamics, for which fluids are a strong representative case. As the reviewer noted, the promising generalization to unseen domains (cloth, sand, smoke) suggests the model is learning a fundamental understanding of how particle-based aggregates move, not just memorizing fluid properties.
>
> We acknowledge that we do not exhaust the full spectrum of intuitive physics (e.g., rigid body collisions or thermodynamics). Our reasoning for this focus is that fluid has always been proposed to be a difficult ([1]Zheng et al., 2024) but foundational material to start with (e.g., robotics manipulation tasks, [2]Lin et al., 2020).
>
> We completely agree that future work should develop more comprehensive benchmarks that span a wider array of physical phenomena to fully assess and cultivate a holistic physical understanding in MLLMs. We will state this clearly in our revised conclusion and discussion.
>
> [1]Zheng, Z., Yan, X., Chen, Z., Wang, J., Lim, Q.Z., Tenenbaum, J.B., & Gan, C. (2024). ContPhy: Continuum Physical Concept Learning and Reasoning from Videos. *ArXiv, abs/2402.06119*.
>
> [2]Lin, X., Wang, Y., Olkin, J., & Held, D. (2020). SoftGym: Benchmarking Deep Reinforcement Learning for Deformable Object Manipulation. *Conference on Robot Learning*.

---

> > ### Author Response · Authors · 2025-11-20
> > **Rebuttal to Reviewer HQBD (Part 2/3)**
> >
> > ### Q2. Experimental Design and Baselines
> >
> > We thank the reviewer for suggesting expanding our comparison to include larger baseline models.
> >
> > - Our paper already provides a scaling analysis in **Table 2 (left panel)** (Qwen2-VL 3B-72B, InternVL2.5 2B-26B), which shows that simply scaling parameters yields diminishing returns and motivates our targeted SDF training.
> >
> > - To further strengthen this point, **we add new baseline results from the recently released Qwen3-VL series** (with varying scales). Our results show that Qwen3VL series are more advanced, but these models still show a clear performance gap on our low-level tasks, reinforcing the need for our method.
> >
> >   | Qwen3VL | 2B   | 4B   | 8B   | 32B  | 235A22B |
> >   | ------- | ---- | ---- | ---- | ---- | ------- |
> >   | NFS     | 26.8 | 29.1 | 27.6 | 30.5 | 33.9    |
> >   | TCV     | 59.1 | 60.2 | 60.9 | 61.1 | 63.6    |
> >
> >
> >
> > ### Q3. Dependence on Synthetic Data and Real-World Applicability
> >
> > The reviewer raises an excellent point about the sim-to-real gap. We would like to clarify a critical aspect of our design: Our testing benchmark is not just synthetic. As described in Section 3.3, it is a challenging mix of synthetic videos *and* **real-world videos** from the web. The strong performance of our SDF-trained model (Figure 4) on this mixed benchmark demonstrates that our method, *trained* on clean simulator data, already generalizes effectively to noisy, real-world video.

---

> > > ### Author Response · Authors · 2025-11-20
> > > **Rebuttal to Reviewer HQBD (Part 3/3)**
> > >
> > > ### Q4. More In-Depth Analysis of MLLM Limitations
> > >
> > > We thank the reviewer for really insightful question which we have tried to investigate. Determining whether the limitations of MLLMs in physical reasoning stem from architectural constraints or data deficiencies is indeed a critical issue. Our initial investigations want to figure out the architectural aspect, we conducted a targeted experiment. We train an architecture similar to V-JEPA with the encoder representation on our dataset to specifically learn to extract low level physical dynamics. However, we found that simply leveraging the encoder representation was not effective. We have included a detailed report of this experiment in our supplementary material (page 24, highlighted in blue).
> > >
> > > This finding led us to focus on the data centric perspective. The recent improvements in models like Qwen3-VL, which was trained on more embodied and spatial task data, lend strong support to this view. Its performance improved on our intuitive physical understanding tasks with the help of these spatial and physical data. It indicates that the underlying architectures of MLLMs are capable of learning physical reasoning, but they must be exposed to the right kind of data. We believe that intuitive physical understanding is a foundational capability. Therefore, attempting to learn it implicitly from high level, end to end task data (such as embodied action) would be akin to putting the cart before the horse. A more effective strategy is to first equip MLLMs with this fundamental capability using targeted data.
> > >
> > > Therefore, our work adopts a data centric approach precisely to address this gap. By generating structured data focused on physical dynamics (SDF), we aim to directly enhance the MLLM's capacity for intuitive physical reasoning, providing a solid foundation for a wide range of real world applications.
> > >
> > >
> > >
> > > We thank the reviewer again for their thoughtful suggestions. We will update the manuscript with the new Qwen3-VL baselines and failure case analysis, all of which strengthen the paper. We are happy to discuss these points further.

---

### Official Review · Reviewer_Vi2b · 2025-10-30

**Soundness:** 2
**Presentation:** 2
**Contribution:** 2
**Rating:** 2
**Confidence:** 4

**Summary:**

The paper proposes two low level tests for intuitive physics in MLLMs, Next Frame Selection and Temporal Coherence Verification, focused mainly on fluids. It introduces Scene Dynamic Field, a simulator derived motion visualization used as a visual prompt within a multi task fine tuning scheme, and reports sizable gains on the proposed tests with some transfer to cloth, sand, smoke, and plasticine.

**Strengths:**

- Clear problem decomposition toward low level dynamics rather than high level QA
- Simple intermediate representation that is easy to plug into existing MLLMs
- Ablations on stride, prompts, model scale, and expert vs self distilled data
- Some transfer beyond fluids and an attention analysis that supports the claim that SDF shifts focus to earlier frames

**Weaknesses:**

- The weakest point is that there is no comparison to strong motion baselines. SDF is a velocity magnitude style visual prompt, which is conceptually close to optical flow magnitude, flow stacks, dynamic images, or even simple frame differencing. Without head to head baselines under the same training data and budget, the gains could come from adding any explicit motion cue rather than from SDF itself. This leaves the central claim unproven.

- Benchmarks are author designed and multiple choice, so improvements may reflect distractor design rather than genuine physics understanding

- Absolute accuracy remains low, so practical impact is unclear

- Limited evaluation beyond fluids for rigid body scenes or causal reasoning tasks, so the title and claims feel broader than what is shown
- Distractor pruning uses SigLIP embeddings which are related to encoders used by evaluated models, creating a risk of bias

My recommendation is reject. The idea is interesting and the empirical gains are clear on the authors benchmark, but the evaluation misses strong motion baselines, relies on potentially biased distractor construction, and the absolute performance and scope do not yet support the paper’s broad claims.

**Questions:**

- How does SDF compare to simple optical flow overlays, grayscale motion magnitude, or event frame stacks when training with the same budget?

- Are results robust when distractors are generated with a feature space disjoint from any model under test?

- Can you evaluate on external physics benchmarks without re curating the data to validate generality?

---

> ### Author Response · Authors · 2025-11-20
> **Rebuttal to Reviewer Vi2b (Part 1/2)**
>
> We sincerely appreciate the reviewer for the insightful feedback and constructive suggestions. We appreciate your positive observations for "our clear problem decomposition toward low-level dynamics" and "evidence of transfer beyond fluids". We have since run additional experiments that further support and validate our original claims.
>
> ### Q1: Comparison to Other Representation Baselines
>
> We thank the reviewer for pointing out the problem of comparing with other representations. We respectfully argue that reconstructing representations from videos such as optical flow is unnecessary for our setting. Our generated simulator provides precise velocity for each particle at every time step. This velocity is a natural source for the Scene Dynamic Field and yields a clean and physically grounded visual prompt for training. In contrast, reconstruction from real world videos is noisy due to occlusion, texture aliasing, and illumination changes, which weakens the learning signal for intuitive physics.
>
> To directly address this concern, we conducted an additional comparison across four prompt conditions using the same question answering protocol:
>
> (1) **w/o Visual Prompt**: directly QA without any visual prompt.
>
> (2) **w/ Optical Flow**: add the reconstructed optical flow in prompt for QA.
>
> (3) **w/ Depth**: add depth estimation of the last frame in prompt for QA.
>
> (4) **w/ SDF(ours)**: add our proposed SDF as visual prompt for QA.
>
> | Visual Prompt | NFS Acc | TCV Acc |
> | ------------- | ------- | ------- |
> | No            |  29.1   |  57.5   |
> | Optical Flow  |  27.0   |  65.1   |
> | Depth         |  32.4   |  60.1   |
> | SDF (ours)    |  41.2   |  68.9   |
>
> These results indicate that optical flow provides some helpful motion cue for TCV but it is less effective on NFS where reconstruction noise limits its discriminative value. In contrast, SDF achieves the best performance across both tasks, showing that clean velocity from the simulator yields a more reliable and more generalizable visual prompt. We include the experiment and an example figure in the revised supplementary on page 16 with blue highlight.
>
> ### Q2: Benchmark Design & Distractor Bias
>
> Thanks for pointing out the potential concern about "potential bias" from SigLIP. We used SigLIP (Section 3.2) precisely to increase the benchmark's difficulty by pruning distractors that were *too easy* (i.e., too dissimilar). To put this concern to rest, we re-generated a test set portion using distractors selected with a disjoint DINOv2 feature space. We found the performance remains similar, there’s little statistical difference for the distractor embedding choice.
>
> | Embedding Model | Tested Model | NFS Acc | TCV Acc |
> | --------------- | ------------ | ------- | ------- |
> | SigLIP          | Qwen2-VL     | 26.8    | 53.2    |
> | DINOv2          | Qwen2-VL     | 25.9    | 53.1    |
> | SigLIP          | InternVL3    | 18.3    | 54.9    |
> | DINOv2          | InternVL3    | 18.0    | 53.1    |
> | SigLIP          | GLM-4.1V     | 25.4    | 54.0    |
> | DINOv2          | GLM-4.1V     | 26.0    | 55.5    |

---

> > ### Author Response · Authors · 2025-11-20
> > **Rebuttal to Reviewer Vi2b (Part 2/2)**
> >
> > ### Q3: Scope & Evaluation on External Benchmarks
> >
> > We thank the reviewer for raising two points here, one on scope and one on external benchmarks.
> >
> > - **On Scope:** Our focus on particle-based continuum dynamics is intentional. Fluids are a notoriously difficult ([1]Zheng et al., 2024) but foundational starting point ([2]Lin et al., 2020). Our transfer experiments (Fig 4) to **cloth, sand, and smoke** already prove the method is not just 'about fluids' but about this general principle.
> >
> >   [1]Zheng, Z., Yan, X., Chen, Z., Wang, J., Lim, Q.Z., Tenenbaum, J.B., & Gan, C. (2024). ContPhy: Continuum Physical Concept Learning and Reasoning from Videos. *ArXiv, abs/2402.06119*.
> >
> >   [2]Lin, X., Wang, Y., Olkin, J., & Held, D. (2020). SoftGym: Benchmarking Deep Reinforcement Learning for Deformable Object Manipulation. *Conference on Robot Learning*.
> >
> > - **On External Benchmarks:** The reviewer's request for evaluation on external benchmarks brings up a core premise of our paper. As stated in our introduction (lines 049-052), high-level reasoning benchmarks (like ContPhy) "entangle multiple capabilities" (vision, language, logic, *and* physics). MLLMs fail at these tasks, but it's impossible to know *why*. Our paper's main contribution is to **disentangle** this problem and be the first to "focus on its very first step... intuitive physics understanding." Our NFS and TCV tasks are specifically designed to isolate and evaluate this fundamental capability, which is a critical and overlooked bottleneck.  We evaluate our method on the high-level **ContPhy** benchmark. The results are shown in the table below:
> >
> > | Type      | Settings | Zero-shot | Finetune | SDF-Ours  |
> > | --------- | -------- | --------- | -------- | --------- |
> > | **Fluid** | Prop.    | 48.00     | 60.00    | 67.06     |
> > |           | Pred.    | 5.77      | 6.73     | 10.38     |
> > | **Soft**  | Prop.    | 50.67     | 42.66    | 50.33     |
> > |           | Pred.    | 21.59     | 40.90    | 42.05     |
> > | **Cloth** | Prop.    | 48.00     | 44.00    | 46.67     |
> > |           | Pred.    | 44.34     | 51.13    | 58.37     |
> >
> > As the table shows, our SDF-Ours model clearly improves prediction performance (e.g., fluid *prediction* from **5.77%** to **10.38%**; cloth *prediction* from **44.34%** to **58.37%**), while the property accuracy improvement is less significant. This is because our proposed SDF does not solve the high-level task outright. This is reasonable and expected: our method is designed to enhance the *first step* (perception). Fully solving these complex reasoning tasks would require incorporating more explicit high-level reasoning modules, which is an important, but distinct, future research direction. Our work provides the essential perceptual foundation that these future reasoning models will require.
> >
> >
> >
> > In summary, our new comparisons against other baseline (e.g., optical flow) and our bias analysis with DINOv2 confirm our original claims. We hope this new evidence makes our contribution clear and resolves the reviewer's concerns.

---

> ### Comment · Reviewer_Vi2b · 2025-11-25
>
> The new experiments clarify several aspects (esp. the SigLIP/DINOv2 distractor issue and the transfer to ContPhy), but the main technical concern that led to my original “reject” remains essentially unaddressed.
>
> 1. Still no matched-baseline for “any explicit motion cue” vs. SDF
>
> My core question was:
>
> > How does SDF compare to simple optical flow overlays, grayscale motion magnitude, or event-frame stacks **when training with the same budget**?
>
> Your rebuttal and Appendix B.6 add Table 7 (“Different Representations”), but that experiment does not answer this:
>
> - In Table 7, optical flow and depth are only **added as extra images at inference time** for QA on a subset of tasks. The model has not been **fine-tuned** to interpret them via your multi-task SDF training pipeline.
> - SDF, in contrast, has been used throughout training in carefully designed dynamic-perception and SDF tasks, the model is **explicitly trained** to exploit that representation. The comparison is therefore between:
>   - “Representation that the model was trained on” vs.
>   - “Representations the model never saw during training.”
> - The optical flow you use is also reconstructed from real videos, whereas SDF is derived from simulator ground-truth velocities. So the comparison conflates at least three factors: (i) representation design, (ii) whether the model was trained to use that representation, and (iii) whether the representation comes from clean simulator signals or noisy video estimation.
>
> This means the new experiment does not isolate whether your gains come from:
>
> 1. The specific SDF encoding (line-integrated blue-channel mapping), **or simply**
> 2. Giving the model **any** explicit motion channel derived from a simulator, together with your multi-task CoT training.
>
> To substantiate the central claim that “Scene Dynamic Field unlocks intuitive physics understanding”, you would need something closer to:
>
> - Use the **same simulated data and velocities** you already have, and derive multiple visual encodings from them, e.g.:
>   - plain grayscale velocity magnitude maps,
>   - 2- or 3-channel (u, v, |v||) flow fields,
>   - temporal “dynamic images” / frame-difference magnitude.
> - Run exactly the same multi-task fine-tuning pipeline (dynamic perception + CoT + NFS/TCV) with each representation, under the same data and compute budget.
> - Compare to an SDF variant that uses no special color coding beyond a simple mapping of |v| to intensity.
>
> If all such variants yield similar improvements, then the contribution is “privileged simulator motion supervision helps MLLMs” rather than anything specific to SDF. Right now, the evidence does not distinguish these possibilities, so the central representational claim remains unproven.
>
> 2. Representation ablation is mismatched to the main evaluation
>
> Related to the above, there is a conceptual mismatch:
>
> - In your main experiments, SDF is only available at **training time**. At test time on real NFS/TCV videos, the model sees only RGB frames and the question.
> - In Table 7, SDF/flow/depth are used as **test-time prompts** on a synthetic subset.
>
> So Table 7 evaluates a different setting from your main claim (“sim‑to‑real improvement via SDF as a training signal”), and it does so under conditions that are more favorable to SDF (clean simulator fields) than to optical flow (noisy reconstruction). This makes it hard to use that table to support the paper’s central conclusion.
>
> Appendix H tries to address the “strong motion baseline” question via a V‑JEPA-style setup, but again the comparison is not compelling:
>
> - All tested video encoders are frozen and only a small 3-layer head is trained.
> - Prior work on V‑JEPA, VideoMAE, InternVideo, etc., fine-tunes the encoder or uses larger heads for downstream tasks, so this experiment is not a realistic “strong” video baseline.
> - Only a simple subset of TCV is evaluated and there is no NFS comparison.
>
> As a result, the conclusion that encoder-only video models are inherently weaker than SDF‑trained MLLMs is not supported by strong evidence.
>
> Your new ContPhy results are useful and show some transfer, but they also reinforce that SDF primarily helps low-level prediction (e.g., modest gains in prediction accuracy) and has limited impact on higher-level property questions. That is consistent with your stated focus, but it further underlines that the work currently demonstrates “better short-horizon frame prediction / coherence on fluid-like scenes after simulator-based finetuning” rather than a broader notion of intuitive physics understanding. So this "central claim" in your paper is not well-supproted in my opinion.
>
> The biggest outstanding issue is still the absence of **matched, training-time baselines using alternative motion representations derived from the same simulator data**. Because of this, it remains unclear whether SDF itself is critical, or whether any explicit, simulator-derived motion cue plus multi-task finetuning would yield similar gains.

---

> > ### Author Response · Authors · 2025-11-28
> > **Follow-up Rebuttal to Reviewer Vi2b (Part 1/2)**
> >
> > We thank the reviewer for the follow-up and happy to know our rebuttal has solved some of your concerns. We appreciate the opportunity to clarify and present the results of the training ablation experiments requested.
> >
> > ### 1. Clarification on Previous Inference-Time Comparison
> >
> > We would like to clarify the exact intent of the 'different representations' experiment (Table 7) in our previous response. Table 7 was a **fair comparison** of zero-shot representational efficacy. That particular experiment used representations into the same untrained base model as a means of evaluating which representation provided more intuitively accessible information to the MLLM 'out of the box'.
> >
> > Most importantly, this comparison proved that betting on easier motion representations reconstructed from real data does not work. This inspires our core approach: we must leverage **simulated data** since only the simulation can provide the **physical ground truth** necessary to supervise the learning of complex dynamics. Regarding a more detailed ablation on how to organize these physical data into our training pipeline, we are working on it because it requires extensive extra effort, and below is our progress.
> >
> > ### 2. New Matched-Training Baselines
> >
> > We fully appreciate the reviewer's challenge to compare representations under the same training regime. We wish to highlight that establishing these matched baselines is a massive work. It requires a complete re-execution of our data pipeline: 1) curating and regenerating training data from the simulator for each representation, 2) rendering specific modalities for training samples, 3) organizing training data and 4) conducting fine-tuning and evaluation.
> >
> > While it is practically impossible to exhaustively evaluate every conceivable motion representation, we have committed all we could to benchmark the two most critical baselines requested: Grayscale Velocity which shows the value of speed while discarding the direction and the frame differences and Frame Difference Heatmap, which visualize the difference between two frames.
> >
> > We respectfully note that a standard 2- or 3-channel flow map (e.g., encoding u, v, |v| directly to RGB) is fundamentally not suitable for **volumetric fluid particle data**. Unlike opaque surface rendering where one pixel corresponds to exactly one surface point, our fluid domain allows many particles to occupy the same line of sight at different depths. A direct projection forces a nearest neighbor or averaging approach, both of which are destructive: the former creates noise by discarding internal structures, while the latter loses distinct flow features through destructive interference. Our **Scene Dynamic Field (SDF)** tries to avoid this by treating the flow as a medium. By strictly integrating density-weighted velocity along the ray (Eq. 6), we preserve the cumulative signal of the fluid volume to capture the dynamics, rather than flattening complex 3D behavior into an ambiguous 2D planar projection.

---

> > > ### Author Response · Authors · 2025-11-28
> > > **Follow-up Rebuttal to Reviewer Vi2b (Part 2/2)**
> > >
> > > **Experimental Setup:** We used the exact same simulator data to train three distinct models based on Qwen2.5VL on NFS task. The only difference was the visual prompt representation derived from the simulator particles:
> > >
> > > 1. **w/o Visual Prompt (Trained)**: Training with the multi-task framework without visual prompt as baseline.
> > > 2. **Velocity (Trained):** Velocity magnitude $|v|$ mapped to grayscale intensity with multi-task framework.
> > > 3. **Difference (Trained):** Heatmap to show the frame-level different with multi-task framework.
> > > 4. **SDF (Ours, Trained):** Our proposed Scene Dynamic Field with multi-task framework.
> > >
> > > The trained model's performance on the test set with different number of training samples are as follows:
> > >
> > > | Training Representation | 500  | 1000 | 2000 |
> > > | ----------------------- | ---- | ---- | ---- |
> > > | w/o Visual Prompt   | 30.1 | 31.1 | 31.5 |
> > > | Velocity                | 32.5 | 33.8 | 34.0 |
> > > | Difference              | 28.2 | 27.1 | 30.2 |
> > > | **SDF (Ours)**          | 33.7 | 35.1 | 38.9 |
> > >
> > > The results clearly compare the utility of different motion signals for physics understanding in VLMs. While explicitly providing velocity magnitude (Velocity) improves performance over the baseline (e.g., +2.4% at 500 samples), it consistently lags behind our Scene Dynamic Field (SDF). This confirms that while scalar speed is a useful signal, it is insufficient on its own without the directional information captured by SDF. Furthermore, our method demonstrates better efficiency: as the training set grows, the gain provided by SDF increases more, indicating that SDF provides a richer signal that models leverage to generalize better with more data.
> > >
> > > Conversely, the Frame Difference representation performs worse. This suggests that simple pixel-level differences introduce more noise or ambiguouty that distract the VLM from the underlying physical dynamics rather than clarifying them. This failure validates our hypothesis that raw visual differences are poor proxies for complex fluid physics compared to physics-informed representations.
> > >
> > >
> > >
> > > ### 3. Clarification on Appendix H
> > > We would like to clarify regarding the purpose of Appendix H. This section was not intended to present a "strong motion baseline" for competitive benchmarking against video models. It was included to respond a specific question from **Reviewer HQBD**, who suggested for an "In-Depth Analysis of MLLM Limitations."
> > >
> > > The experiment appended in H was targeted at probing whether the lags of MLLMs in physical reasoning are due to architectural constraints or data deficiencies, under a controlled V-JEPA-style study with a frozen encoder, rather than its capacity to be fine-tuned. From the results obtained, we derive that mere leveraging of existing encoder representations is not to any great extent effective for understanding physical dynamics, which supports our datacentric hypothesis: MLLMs should be exposure to targeted, physics-rich data (like our SDF) rather than implicit learning from high-level tasks.

---

### Official Review · Reviewer_deha · 2025-11-01

**Soundness:** 4
**Presentation:** 3
**Contribution:** 3
**Rating:** 8
**Confidence:** 3

**Summary:**

The paper focuses on the task of fluid and physical scene understanding. It makes two primary contributions:

1. The authors build a comprehensive benchmark for physical reasoning from videos, introducing two key tasks — Next Frame Selection (NFS) and Temporal Coherence Verification (TCV). The dataset is constructed from multiple sources, including ContPhy, PhysBench, and real-world video clips, to cover both synthetic and natural dynamics.

2. The paper proposes the Scene Dynamic Field, a framework that leverages physical simulation principles and integrates Chain-of-Thought (CoT) reasoning with explicit physics modeling. This approach enables multimodal models to reason about scene dynamics beyond static appearance, bridging the gap between perception and physical understanding.

**Strengths:**

The paper tackles an important and challenging problem — physical and fluid scene understanding — which is a key step toward enabling models to reason about real-world dynamics.

The proposed explicit physical reasoning task is interesting and well-motivated, encouraging models to go beyond visual perception and engage in physically grounded understanding.

The paper is generally well-written and structured, with a clear motivation and logical flow from problem definition to methodology and experiments.

**Weaknesses:**

1. Limited physical understanding of only fluid dynamics. The proposed framework primarily focuses on liquid and fluid dynamics, which narrows the generality of the method. Extending the benchmark and tasks to cover a broader range of physical interactions — such as rigid-body motion, collision, or elastic deformation — would strengthen the overall contribution.

2. Few missing references to physical reasoning VQA. The paper could benefit from citing and discussing more recent physical understanding and VQA-related research, such as Comphy (https://arxiv.org/abs/2205.01089) and DynSuperCLEVR (https://arxiv.org/abs/2406.00622), which address the VQA for physical reasoning.

**Questions:**

1.  Data Modality and Encoder Usage:
What is the exact data format of the SDF samples used for training? From Figure 3, the SDF appears to be an RGB-like image generated through velocity-to-color mapping. Could the authors clarify whether these SDF images are indeed three-channel RGB inputs, and if they are processed by the same image encoder as the regular video frames during training?

2. Ablation on the SDF Step in CoT Fine-Tuning. In the SDF-guided Chain-of-Thought fine-tuning process, the framework first predicts the SDF representation and then predicts the next frame based on that SDF. What would happen if we retain the same training pipeline but skip the explicit SDF step, i.e., directly fine-tune the model with simplified CoT instructions to predict the RGB frame without generating the SDF?
This comparison would help clarify how much the explicit SDF stage contributes to physical reasoning, especially since the SDF image appears visually similar to the original frame except for color-coded fluid regions (as shown in Figure 3). Such an ablation could reveal whether the SDF acts mainly as a visual prompt or as a truly distinct physical representation.

---

> ### Author Response · Authors · 2025-11-20
> **Rebuttal to Reviewer deha (Part 1/2)**
>
> We are very grateful to the reviewer for their positive comments that our paper tackles an "important and challenging problem," and that our "explicit physical reasoning task is interesting and well-motivated". We will address the reviewer's constructive suggestions below.
>
> ### W1. Physical understanding of only particle-based dynamics.
>
> We thank the reviewer for this point. We acknowledge that our primary benchmark and training data focus on particle-based dynamics. However, our core hypothesis is that our method teaches the model a more fundamental understanding of **particle-based continuum dynamics**, for which fluids are a strong, representative example which has been chosen as training data. To test this, we included transfer experiments (Section 4.3) that evaluate the model on unseen domains like **cloth, sand, and smoke**.
>
> The strong positive results in these transfer tasks demonstrate that the model is not just memorizing fluid properties but is learning a generalizable, intuitive understanding of how particle-based aggregates move and interact.
>
> We agree that extending this work to rigid-body collisions or elastic deformations is an excellent direction for future work, and we will add this to our discussion part.
>
> ### W2. References to physical reasoning VQA.
>
> We thank the reviewer for pointing out these relevant papers. `Comphy` and `DynSuperCLEVR` are indeed important related works. We will add a discussion of both to our Related Work (Section 2) to provide a more comprehensive overview of the physical reasoning VQA landscape.

---

> > ### Author Response · Authors · 2025-11-20
> > **Rebuttal to Reviewer deha (Part 2/2)**
> >
> > ### Q1. Data Modality and Encoder Usage
> >
> > This is an excellent question that gets to a key part of our design. T**he SDF is a three-channel RGB image** processed by the same image encoder as the regular video frames. However, the "velocity-to-color mapping" (described in Section 4.1) is **only applied to the fluid particles themselves**. Critically, the surrounding static background remains unchanged and is identical to the original RGB frame.
> >
> > This design is deliberate. By keeping the background static, we force the model to learn by comparison. It can "ground" the new, dynamic-colored fluid regions against the familiar, static context of the scene (e.g., the cup, the table). This teaches the model to associate the blue-tinted visual prompt *specifically* with the motion of the fluid, rather than treating it as an entirely separate, abstract image.
> >
> > ### Q2. Ablation on the SDF Step in CoT Fine-Tuning
> >
> > This is a good suggestion that our "CoT" (Reasoning Only) setting in Figure 4A serves as a *zero-shot* baseline for this, but a more detailed comparison with a trained model is needed.
> >
> > To address this, we run an additional ablation study for this. We train the Qwen2VL model using the exact same multi-task fine-tuning pipeline, but with the SDF-related tasks remove the SDF representation. This model will be fine-tuned *only* on the original NFS/TCV tasks with CoT instructions.
> >
> > We use parameter efficient LoRA to train the CoT w/o SDF and w/ SDF to show the results. The performance drop for CoT training w/o SDF demonstrates that the visual SDF representation is indeed critical for the model to learn the intuitive physical understanding.
> >
> > | Settings                    | NFS Acc | TCV Acc |
> > | --------------------------- | ------- | ------- |
> > | Zero-shot                   | 26.8    | 53.2    |
> > | CoT Prompt                  | 29.8    | 60.1    |
> > | CoT training w/o SDF      | 33.0    | 55.5    |
> > | CoT training w/ SDF (ours) | 38.9    | 66.1    |
> >
> > In summary, we again thank the reviewer for their valuable and constructive feedback. We will update the revised manuscript to include the suggested related works (W2) and the new ablation study on the CoT pipeline (Q2). We will also clarify the scope of our work on particle-based dynamics and its successful generalization (W1). Thanks again for providing suggestions on these additions which will strengthen the paper.

---

### Official Review · Reviewer_tuv4 · 2025-11-02

**Soundness:** 3
**Presentation:** 3
**Contribution:** 3
**Rating:** 6
**Confidence:** 3

**Summary:**

This paper try to addresses the gap in MLLMs understanding of intuitive physics, particularly for continuum objects like fluids. The authors first introduce two low-level benchmark tasks, next frame selection and temporal coherence verification, to demonstrate that current MLLMs perform poorly at perceiving physical dynamics. To solve this, they propose SDF, an intermediate representation generated by physics simulators that visually encodes motion (e.g., velocity as color intensity). Through a multi-task fine-tuning strategy, their SDF-enhanced model achieves substantial gains on fluid tasks and shows strong generalization to unseen physical domains like cloth and sand.

**Strengths:**

- The motivation of the paper correctly focuses on a simpler, core problem, which is just perceiving physical motion, separating it from complex, high-level reasoning.

- Using a visual map (SDF) from a physics simulator to train the model works. This helps the model learn the idea of dynamics, letting it generalize from fluids to unseen materials like cloth or sand.

- The authors built a strong benchmark by mixing simulated data with real-world videos, and they ran thorough experiments to validate their approach.

- The paper is well presented with clear logic.

**Weaknesses:**

- The SDF representation is very basic, encoding just the projected velocity magnitude into one color channel. It's questionable if this simple map captures enough information for complex interactions, or if it needs to include more data like 3D vector direction or using optical flow (which avoids physics simulation). A discussion comparing the performance, advantages, and disadvantages of these different representations would greatly strengthen the analysis.

- The pipeline is complex. It requires a full-parameter fine-tuning process, data from simulators, and knowledge distilled from expert models, which is computationally expensive and hard to scale.

**Questions:**

See in the weekness.

---

> ### Author Response · Authors · 2025-11-20
> **Rebuttal to Reviewer tuv4 (Part 1/2)**
>
> We sincerely thank the reviewer for their insightful feedback and constructive suggestions. We appreciate the reviewer's positive feedback that our motivation "correctly focuses on a simpler, core problem" by separating physical motion perception from complex reasoning. We are also glad the reviewer found that our representation SDF "helps the model learn the idea of dynamics, letting it generalize." We have provided detailed responses below to address the reviewer's questions and concerns.
>
> ### Q1:  **Regarding the simplicity of the SDF representation and comparison to alternatives:**
>
> The reviewer correctly observes that our Scene Dynamic Field (SDF) is a concise representation, encoding projected velocity magnitude into a single color channel. This was a deliberate design choice and is central to our paper's core objective: to train Vision-Language Models (VLMs) for intuitive physical understanding.
>
> 1. **The Goal is Abstract Intuitive Understanding, Not Fine-Grained Reconstruction:** Our primary goal is to enhance the *intuitive physics understanding* of MLLMs for high-level reasoning and planning. As we state in Section 5, we aim to "appropriately abstract"  representations from simulators. The SDF is designed to be an intermediate visual prompt that is "perceptually calibrated" and optimized for VLM consumption, rather than a high-fidelity, complex data field.
>
> 2. **VLM Suitability and Scalability:** As the reviewer suggests, representations like full 3D velocity vectors would contain more data. However, we argue that current VLMs are not well-equipped to directly interpret such fine-grained, raw vector fields. This would require massive, specialized training for the VLM to develop the necessary 3D spatial understanding, which runs counter to our goal of a "cost-efficient approach". 3D vectors could be challenging for current VLMs to leverage directly. As noted in recent work (e.g., [1]Fan et al., 2025; [2] Zheng et al., 2024), enabling VLMs to gain a true 3D spatial and vector-based understanding typically requires massive-scale, specialized effort. Our focus, in contrast, is to inject intuitive physical dynamics so that a VLM can readily consume and integrate into its reasoning process, making our abstract representation a more suitable choice for this goal.
>
>    [1] Fan, Zhiwen et al. VLM-3R: Vision-Language Models Augmented with Instruction-Aligned 3D Reconstruction. *ArXiv* abs/2505.20279 (2025)
>
>    [2] Zheng, D., Huang, S., & Wang, L. (2024). Video-3D LLM: Learning Position-Aware Video Representation for 3D Scene Understanding. CVPR 2025
>
> 3. **On Alternative Representations (e.g., Optical Flow):** To directly address the reviewer's excellent suggestion, we conducted an additional experiment comparing our SDF to other representations. We directly evaluate SDF's effectiveness by comparing across four prompt conditions using the same question answering protocol:
>
> (1) **w/o Visual Prompt**: directly QA without any visual prompt.
>
> (2) **w/ Optical Flow**: add the reconstructed optical flow in prompt for QA.
>
> (3) **w/ Depth**: add depth estimation of the last frame in prompt for QA.
>
> (4) **w/ SDF(ours)**: add our proposed SDF as visual prompt for QA.
>
> | Visual Prompt | NFS Acc | TCV Acc |
> | ------------- | ------- | ------- |
> | No            |  29.1   |  57.5   |
> | Optical Flow  |  27.0   |  65.1   |
> | Depth         |  32.4   |  60.1   |
> | SDF (ours)    |  41.2   |  68.9   |
>
> These results indicate that optical flow is less effective on NFS where reconstruction noise limits its discriminative value. In contrast, SDF achieves the best performance across both tasks, showing that clean velocity from the simulator yields a more reliable and more generalizable visual prompt.
>
> Our simulator-generated SDF, while simpler, provides a *cleaner* and *more reliable* signal of the underlying motion. It bypasses the noise of 2D reconstruction and directly encodes the "ground truth" dynamics from the physics engine. This reliability is crucial for building a robust intuitive understanding. We include the experiment and an example figure in the revised supplementary on page 16 with blue highlight. We thank the reviewer again for prompting this valuable addition.

---

> > ### Author Response · Authors · 2025-11-20
> > **Rebuttal to Reviewer tuv4 (Part 2/2)**
> >
> > ### Q2: **Regarding the complexity and scalability of the pipeline:**
> >
> > We thank the reviewer for raising this important point about complexity and cost. However, we respectfully argue that the primary cost bottleneck in this domain is **not** the one-time training compute, but the acquisition of high-quality, physically-grounded **data**. Our method is exceptionally "cost-efficient" from this critical, data-centric perspective.
> >
> > 1. **Cost of Simulator Data vs. Real-World Data:** The alternative by collecting, segmenting, and accurately annotating the ground-truth physical dynamics of real-world videos (especially for fluids) is extraordinarily expensive, time-consuming, and in many cases, physically impossible. In contrast, our simulation-based approach provides a virtually infinite source of clean, physically-grounded data at a fraction of this cost.
> > 2. **Cost of Distillation:** The same logic applies to the use of expert models for distillation. As our ablation study in the Appendix (Table 6) shows, this is an efficient process, and it represents a scalable, one-time data generation step that is far more practical than any human-in-the-loop alternative.
> > 3. **Scalability via Parameter-Efficient Tuning:** To directly address the reviewer's concern about computational scalability, we also trained a **parameter-efficient (LoRA)** version (rank=8 and alpha=32) of our method. We found it still achieves most of the improvement while drastically reducing the training cost. This demonstrates that our SDF-guided framework is effective and also offers a favorable trade-off for low-resource scenarios.
> >
> > | Model              | # Trainable Para. | NFS Acc | TCV Acc |
> > | ------------------ | ----------------- | ------- | ------- |
> > | Qwen2VL-ZeroShot   | N/A               | 26.8    | 53.2    |
> > | Qwen2-VL-SDF       | 7615.62M          | 41.2    | 70.1    |
> > | Qwen2-VL-SDF(LoRA) | 20.19M            | 38.9    | 66.1    |
> > | GLM4V-ZeroShot     | N/A               | 25.4    | 53.9    |
> > | GLM4V-SDF          | 9399.95M          | 46.1    | 80.2    |
> > | GLM4V-SDF(LoRA)    | 21.18M            | 44.0    | 75.4    |
> >
> > In summary, while our method requires a one-time fine-tuning for better performance, it represents a far more scalable and practical paradigm by solving the primary, and far more difficult, bottleneck of data acquisition. We will add the LoRA results to the revised manuscript to further demonstrate this point.

---

### Meta-Review · Area_Chair_SHBh · 2025-12-18

**Summary:**

Reviewers acknowledged the motivation and potential of the work but raised several concerns about scope, evaluation, and evidence supporting the central claims. A major issue was the lack of strong, matched motion baselines: reviewers repeatedly argued that it was unclear whether the gains stem from the specific Scene Dynamic Field (SDF) representation or simply from providing any explicit, simulator-derived motion cue combined with multi-task fine-tuning. Earlier comparisons were viewed as mismatched (e.g., inference-time prompts vs. training-time supervision, noisy optical flow vs. clean simulator signals), making it difficult to isolate the contribution of SDF itself. Reviewers also questioned the breadth of the claims, noting that experiments focus primarily on fluids and particle-based continuum dynamics, with limited evidence for rigid-body interactions, collisions, or higher-level causal reasoning. Concerns were raised about benchmark design, including reliance on author-curated multiple-choice tasks, potential distractor bias, and low absolute accuracy, which together cast doubt on real-world impact. Several reviewers had concerns about complexity and scalability, pointing to dependence on simulators, fine-tuning cost, and integration with practical settings. Finally, some reviewers felt that improvements on external benchmarks mainly affected low-level prediction rather than demonstrating a broader notion of intuitive physics understanding, suggesting that the paper’s claims may be stronger than what the current evidence supports.

**Reviewer Concerns:**

The rebuttal convincingly addressed several technical and experimental concerns raised by the reviewers. Most notably, the authors responded to repeated requests for motion-representation baselines by adding new, matched training-time ablations using the same simulator data and multi-task pipeline (e.g., grayscale velocity magnitude and frame-difference heatmaps), showing that SDF consistently outperforms these alternatives as training data scales. This directly responds to criticisms from Reviewer Vi2b about conflating representation quality with training exposure. The authors also clarified implementation details around SDF (RGB format, shared encoder, background preservation) and provided requested ablations removing the SDF step, demonstrating a clear performance drop and strengthening the claim that SDF is a critical component rather than incidental. Concerns about distractor bias were addressed by regenerating test sets with a disjoint embedding space (DINOv2), yielding similar results. In addition, the rebuttal expanded baseline coverage to newer models (e.g., Qwen3-VL), added LoRA-based results to mitigate scalability concerns, and included external benchmark results (ContPhy) that show consistent gains in prediction-related subtasks.

Despite these improvements, some substantive issues remain only partially resolved. While the new ablations strengthen the case for SDF over simpler motion cues, the generality of the representational claim is still debatable: the strongest evidence is limited to a small set of alternative encodings (velocity magnitude and frame differences), and it remains unclear whether other physics-informed but non-SDF encodings derived from the same simulator (e.g., richer multi-channel fields) would narrow the gap. Reviewers’ concerns about the breadth of intuitive physics understanding also largely persist. The empirical gains are strongest for low-level, short-horizon prediction and temporal coherence, with limited improvement on higher-level property reasoning; this makes the paper’s broader framing of “unlocking intuitive physics understanding” feel stronger than what is conclusively demonstrated. Additionally, although the authors clarified and defended their benchmark design, skepticism remains about real-world impact, given the continued low absolute accuracies and reliance on author-designed multiple-choice tasks.

**Reviewer Scores:**

Reviewer tuv4 (score 6) was concerned about the simplicity of the SDF representation, the complexity and scalability of the pipeline, and whether alternative motion cues (e.g., optical flow) might suffice. The rebuttal directly addressed these points with (i) new comparisons to optical flow and depth, (ii) LoRA-based parameter-efficient fine-tuning results, and (iii) a clearer data-centric argument about simulator cost versus real-world annotation. The additional evidence likely strengthens the confidence of the reviewer but does not fundamentally change the original position.

Reviewer deha (score 8) was largely positive and focused on clarifications and scope, requesting ablations on the SDF step, encoder usage details, and broader discussion of physical reasoning literature. The rebuttal fully addressed these requests with clean ablations, explicit encoder clarification, added related work, and transfer results beyond fluids. The score would likely remain 8, possibly with increased confidence.

Reviewer Vi2b (score 2) raised the most serious and technically detailed concerns, primarily about missing training-time motion baselines, benchmark bias, and overstated claims. Over the course of the discussion, the authors eventually provided exactly the kind of matched ablations Vi2b requested (training with grayscale velocity and frame-difference representations derived from the same simulator data and pipeline). However, Vi2b’s follow-up comments indicate persistent concerns - even after partial clarifications, the reviewer explicitly stated that the “central representational claim remains unproven” and narrowed the contribution to short-horizon prediction rather than broad intuitive physics. Given this trajectory, it is unlikely that Vi2b would substantially revise the initial position, though the new matched-baseline results might soften the concerns and lead to an increase in the score to 4.

Reviewer HQBD (score 6) was generally supportive but cautious, emphasizing limited scope, dependence on synthetic data, baseline breadth, and the need for deeper analysis of MLLM limitations. The rebuttal directly addressed these issues by adding larger model baselines (Qwen3-VL), clarifying sim-to-real evaluation using mixed real/synthetic benchmarks, and including an architectural vs. data-centric analysis (Appendix H). These additions align well with HQBD’s requests. Since HQBD already leaned toward acceptance but expressed reservations, the score would likely remain 6.

The AC shares the generally positive view of the paper, especially after the rebuttal addressed many of the valid concerns raised by the reviewers. The main concern raised by Reviewer Vi2b was substantially addressed through the addition of matched training-time baselines using velocity encoded as a grayscale image. That said, I agree that a direct comparison against clean optical flow extracted from the simulator and used at training time would further strengthen the empirical evidence. While this does not appear critical for acceptance given the current results, such a comparison would be a valuable addition to the camera-ready version.

---

### Decision · Program_Chairs · 2026-01-26

Accept (Poster)